# On the Sample Complexity of Learning under Invariance and Geometric Stability

**Alberto Bietti**
NYU[*]
alberto.bietti@nyu.edu

**Luca Venturi**
NYU[†]
lv800@nyu.edu

**Joan Bruna**
NYU[‡]
bruna@cims.nyu.edu

## Abstract

Many supervised learning problems involve high-dimensional data such as images, text, or graphs. In order to make efficient use of data, it is often useful to leverage certain geometric priors in the problem at hand, such as invariance to translations, permutation subgroups, or stability to small deformations. We study the sample complexity of learning problems where the target function presents such invariance and stability properties, by considering spherical harmonic decompositions of such functions on the sphere. We provide non-parametric rates of convergence for kernel methods, and show improvements in sample complexity by a factor equal to the size of the group when using an invariant kernel over the group, compared to the corresponding non-invariant kernel. These improvements are valid when the sample size is large enough, with an asymptotic behavior that depends on spectral properties of the group. Finally, these gains are extended beyond invariance groups to also cover geometric stability to small deformations, modeled here as subsets (not necessarily subgroups) of permutations.

## 1 Introduction

Learning from high-dimensional data is known to be statistically intractable without strong assumptions on the problem. A canonical example is learning Lipschitz functions, which generally requires a number of samples exponential in the dimension due to the curse of dimensionality (*e.g.*, [31]). Many high-dimensional machine learning problems involve highly structured data such as images, text, or graphs, and may exhibit invariance to certain transformations of the input data, such as permutations, translations or rotations, and near invariance to small deformations. More precisely, if $\mathcal{X}$ is the high-dimensional data domain, and $G$ is a set of transformations $\sigma : \mathcal{X} \to \mathcal{X}$, the learning task can be alleviated if one knows in advance that the target function $f$ varies smoothly under transformations in $G$: $|f(\sigma \cdot x) - f(x)|$ is uniformly small over $x \in \mathcal{X}$ for $\sigma \in G$.

To further motivate this property, it is useful to view the data domain $\mathcal{X}$ as a space of signals $\mathcal{X} = L^2(\Omega; \mathbb{R})$ defined over a geometric domain $\Omega$, such as a 2d grid. The set of transformations $G$ can then be articulated in terms of $\Omega$ rather than $\mathcal{X}$, a much simpler geometric object, and then *lifted* into $\mathcal{X}$ by composition: if $\sigma : \Omega \to \Omega$, and $x \in \mathcal{X}$ then $(\sigma \cdot x)(u) := x(\sigma^{-1}(u))$ for every $u \in \Omega$. The smoothness to transformations can thus be interpreted as a form of *geometric stability*.

In this paper, we quantify the sample complexity gains brought by geometric stability. Concretely, we consider target functions $f$ defined on the sphere $\mathcal{X} = \mathbb{S}^{d-1}$ in $d$ dimensions with finite $L^2(\mathbb{S}^{d-1})$ norm. In this case, we view the geometric domain as the discrete 1d grid $\Omega = [1, \ldots, d]$, and consider geometric transformations $G$ as subsets of the symmetric group of permutations of $d$

---

[*]Center for Data Science, New York University.

[†]Courant Institute for Mathematical Sciences, New York University.

[‡]Center for Data Science and Courant Institute for Mathematical Sciences, New York University.

35th Conference on Neural Information Processing Systems (NeurIPS 2021).

elements. Given a set $G$ (not necessarily a group), we consider the *smoothing* operator given by $S_G f(x) = \frac{1}{|G|} \sum_{\sigma \in G} f(\sigma \cdot x)$ for $f \in L^2(\mathbb{S}^{d-1})$, and assume that our target function $f$ is geometrically stable, in the sense that $f = S_G g$ for some $g \in L^2(\mathbb{S}^{d-1})$. In words, the smoothing operator $S_G$ replaces the prediction $f(x)$ by the average over transformations of $x$. In particular, functions which are invariant under the action of $\sigma \in G$, namely

$$f(\sigma \cdot x) = f(x), \quad \sigma \in G, x \in \mathbb{S}^{d-1}, \tag{1}$$

are also stable, with $f = S_G f$.

Building on the recent work [23], we proceed by studying harmonic decompositions of such functions using spherical harmonics [14], which generalize Fourier series on the circle to higher dimensions. This allows us to obtain rates of approximation for invariant and geometrically stable functions with varying levels of smoothness, and to study the generalization properties of invariant kernel methods using kernels defined on the sphere. Specifically, our main contributions are:

- By comparing spectral properties of usual kernels on the sphere with invariant ones, we find that the latter provide improvements in sample complexity by a factor of the order of the size of the group when the sample size is large enough (Section 3).

- We study how this improvement factor varies with sample size, in terms of the structure of the group and on spectral properties of the permutation matrices it contains (Section 4).

- We extend the invariance analysis to geometrically stable functions, establishing similar gains in sample complexity that depend on the size of the transformation subset (Section 5).

Our proofs rely on comparing the dimension of invariant and non-invariant spherical harmonics at a given degree, and showing that their ratio decays to the inverse group size as the degree tends to infinity. In contrast to [23], we consider the dimension to be fixed and study non-parametric rates of convergence for potentially non-smooth target functions and general groups of permutations, while they consider a different regime in high dimension and focus on invariance to translation groups.

**Related work.** Invariance and deformation stability have been analysed using convolutional neural network-type architectures such as the scattering transform [22, 7], or convolutional kernels [6, 20]. While these works characterise the stability in terms of the dyadic structure of convolutional filters (such as wavelets), they do not cover a statistical analysis of sample complexity. Similarly, models of compositional functions such as those in [11, 24, 27] study the benefit of hierarchical representations with local connectivity for approximation, while [19, 21] study benefits of local connectivity with optimization-based algorithms; yet these works do not consider invariance or stability. [23] studies similar benefits of invariance but in a different, high-dimensional, regime where only polynomials can be learned, focusing on translation groups, while we consider arbitrary groups or sets of permutations in fixed dimension. [15] also sudies benefits of group invariance, but focuses on linear models, and only considers interpolating estimators. [30] study general generalization bounds of invariant classifiers that scale exponentially with the dimension, which would be pessimistic under our assumptions. [10] study benefits of equivariant kernels in structured prediction problems. [13] studies generalisation advantages of CNNs over fully-connected models, while our focus is on non-parametrics.

## 2 Preliminaries

In this section, we describe our setup and provide some background on harmonic decompositions on the sphere, and how these are affected by invariance.

**Statistical learning setup.** We consider a supervised learning problem where the data distribution $\rho$ on input-label pairs $(x, y)$ is such that $x \in \mathbb{S}^{d-1}$ and $\mathbb{E}[y|x] = f^*(x)$ for some target function $f^*$ in $L^2(\mathbb{S}^{d-1})$. For simplicity, we will assume that $x$ is uniformly distributed on the sphere, and denote the uniform measure on $\mathbb{S}^{d-1}$ by $d\tau$. We consider a regression setup with $L^2$ risk given by

$$R(f) = \mathbb{E}_{(x,y) \sim \rho} \left[ (f(x) - y)^2 \right].$$

For a given estimator $\hat{f}_n$ based on $n$ samples from $\rho$, the goal is then to obtain generalization bounds as a function of $n$ on the excess risk

$$\mathbb{E}[R(\hat{f}_n)] - R(f^*) = \mathbb{E}[\|\hat{f}_n - f^*\|^2_{L^2(d\tau)}], \tag{2}$$

where the expectation is over the $n$ samples. Such bounds are well-studied for various classes of target functions $f^*$ such as smoothness classes, and estimators such as kernel ridge regression. These are typically studied through harmonic decompositions of $f^*$ and of a kernel function in appropriate $L^2$ bases, which then relate function regularity and decays of Fourier coefficients.

**Harmonic analysis on the sphere.** When considering functions in $L^2(d\tau)$, an appropriate choice of orthonormal basis is that of spherical harmonic polynomials [1, 14]. More precisely, denote $\{Y_{k,j}\}_{j=1}^{N(d,k)}$ denote an orthonormal basis of the space $V_{d,k}$ of spherical harmonics of degree $k$, *i.e.*, homogeneous harmonic polynomials of degree $k$, where $N(d,k) = \frac{2k+d-2}{k}\binom{k+d-3}{d-2}$. Then, the collection $\{Y_{k,j} : k \geq 0, j = 1, \ldots, N(d,k)\}$ forms an orthonormal basis of $L^2(d\tau)$, so that any function $f \in L^2(d\tau)$ may be written

$$f(x) = \sum_{k \geq 0} \sum_{j=1}^{N(d,k)} a_{k,j} Y_{k,j}(x), \tag{3}$$

with $\sum_k \sum_{j=1}^{N(d,k)} a_{k,j}^2 < \infty$. Similarly, any dot-product kernel $K(x,x') = \kappa(\langle x, x' \rangle)$ on the sphere may be written

$$\kappa(\langle x, x' \rangle) = \sum_{k \geq 0} \mu_k \sum_{j=1}^{N(d,k)} Y_{k,j}(x) Y_{k,j}(x'), \tag{4}$$

where $\mu_k$ is given by $\mu_k = \frac{\omega_{d-2}}{\omega_{d-1}} \int_{-1}^{1} \kappa(t) P_{d,k}(t)(1-t^2)^{(d-3)/2} dt$. Here, $\omega_{p-1}$ is the surface measure of the sphere in $p$ dimensions, and $P_{d,k}$ are Legendre or Gegenbauer polynomials of degree $k$ in $d$ dimensions (normalized with $P_{d,k}(1) = 1$),which form an orthogonal basis of $L^2([-1,1], dw)$, with $dw(t) = (1-t^2)^{(d-3)/2} dt$. When the kernel $K$ is positive definite and is used in the context of kernel ridge regression with data uniformly distributed on the sphere, then the $\mu_k$ also correspond to the eigenvalues of the covariance operator. These eigenvalues and their decay then control the statistical properties of the kernel ridge regression estimator [8].

**Spherical harmonics and group-invariant functions.** In order to describe harmonic decompositions of functions satisfying the group invariance property (1) for a discrete group $G$, we follow [23] and define the symmetrization operator

$$S_G f(x) = \frac{1}{|G|} \sum_{\sigma \in G} f(\sigma \cdot x). \tag{5}$$

This operator acts as a projection from $L^2(d\tau)$ to a subset thereof which contains invariant functions. It can be shown that the spaces $V_{d,k}$ of spherical harmonics of degree $k$ are stable by $S_G$ [23], and we may then define an orthonormal basis of $\overline{V}_{d,k} := S_G V_{d,k}$ consisting of *invariant* spherical harmonics $\{\overline{Y}_{k,j}\}_{j=1}^{\overline{N}(d,k)}$. We then have the following lemma.

**Lemma 1** (Representation of projection [23]). *For any $k \geq 0$, we have*

$$\gamma_d(k) := \frac{\overline{N}(d,k)}{N(d,k)} = \frac{1}{|G|} \sum_{\sigma \in G} \mathbb{E}_x \left[ P_{d,k}(\langle \sigma \cdot x, x \rangle) \right]. \tag{6}$$

The quantity $\gamma_d(k)$ will play an important role in determining the gains in sample complexity brought by invariance. We will show in Section 4 that $\gamma_d(k)$ converges to $1/|G|$ for large $k$, with an asymptotic behavior that is governed by spectral properties of the elements of the group.

## 3 Sample Complexity of Invariant Kernels

We begin our study by focusing on the invariant case. In this section, we study the sample complexity of learning invariant functions, by considering kernel ridge regression estimators and providing non-parametric rates of convergence that illustrate the gains achievable with invariant kernels compared to non-invariant ones.

**Kernel ridge regression (KRR) and invariant kernels.** For a positive definite kernel $K$ with RKHS $\mathcal{H}_K$, we consider the KRR estimator $\hat{f}_\lambda$ given by

$$\hat{f}_\lambda := \arg\min_{f \in \mathcal{H}_K} \frac{1}{n} \sum_{i=1}^n (f(x_i) - y_i)^2 + \lambda \|f\|_{\mathcal{H}_K}^2. \tag{7}$$

We consider the following kernels, which we assume positive definite, given for $x, x' \in \mathbb{S}^{d-1}$ by

$$K(x, x') = \kappa(\langle x, x' \rangle), \qquad K_G(x, x') = \frac{1}{|G|} \sum_{\sigma \in G} \kappa(\langle \sigma \cdot x, x' \rangle), \tag{8}$$

with $\kappa(u) \leq 1$. A common example for $\kappa$ is the arc-cosine kernel [9], which arises from infinite-width shallow neural networks with ReLU activations. The following integral operator defined on $L^2(d\tau)$ and its eigen decomposition play an important role for the statistical and approximation properties of kernel methods:

$$T_K f(x) = \int K(x, x') f(x') d\tau(x'). \tag{9}$$

We now show that its spectral properties are closely related for $K$ and $K_G$.

**Lemma 2** (Spectral properties of $K$ and $K_G$.). *There exists a basis of spherical harmonics in which the operators $T_K$ and $T_{K_G}$ are jointly diagonalized. They admit the same eigenvalues $\mu_k$ as in* (4)*, with multiplicity $N(d, k)$ for $T_K$ and $\overline{N}(d, k)$ for $T_{K_G}$.*

The decay of the eigenvalues $\mu_k$ controls the smoothness of functions in the RKHS, for instance when $\mu_k$ decays polynomially, $T_K$ behaves similarly to powers of the Laplacian on the sphere, leading to functional spaces similar to Sobolev spaces. For the example of the arc-cosine kernel, $\mu_k$ decays as $k^{-d-2}$. leading to an RKHS containing functions with $d/2 + 1$ bounded derivatives [2].

**Approximation error.** The approximation error of kernel methods is often controlled by the following quantity (*e.g.*, [3, 12]):

$$A_{\mathcal{H}}(\lambda, f^*) = \inf_{f \in \mathcal{H}} \|f - f^*\|_{L^2(d\tau)}^2 + \lambda \|f\|_{\mathcal{H}}^2, \tag{10}$$

where $f^*$ is a target function in $L^2(d\tau)$, and $\mathcal{H}$ is a given RKHS. In particular, if $f^*$ is smooth enough so that $f^* \in \mathcal{H}$, then we have $A_{\mathcal{H}}(\lambda, f^*) \leq \lambda \|f^*\|_{\mathcal{H}}^2$, while if $f^* \notin \mathcal{H}$, *e.g.*, if $f^*$ is only Lipschitz, then $A_{\mathcal{H}}(\lambda, f^*)$ typically grows much faster with $\lambda$. We now show a useful result for invariant targets, showing that in this case the approximation error is the same for the kernels $K$ and $K_G$.

**Lemma 3** (Approximation error for invariant functions.). *If $f^*$ is invariant to the group $G$, so that $f^* = S_G f^*$, then we have*

$$A_{\mathcal{H}_K}(\lambda, f^*) = A_{\mathcal{H}_{K_G}}(\lambda, f^*). \tag{11}$$

**Degrees of freedom.** The above result suggests that any gains of using $K_G$ instead of $K$ for learning invariant functions should come from estimation rather than approximation error. The estimation error of ridge rigression estimators is typically controlled with the following quantity, often called *degrees of freedom* or *effective dimension* (*e.g.*, [3, 18]):

$$\mathcal{N}_K(\lambda) = \text{Tr}(\Sigma_K (\Sigma_K + \lambda I)^{-1}) = \sum_{m \geq 0} \frac{\lambda_m}{\lambda_m + \lambda}, \tag{12}$$

where $\Sigma_K = \mathbb{E}_x [K(x, \cdot) \otimes_{\mathcal{H}_K} K(x, \cdot)]$ is the covariance operator and $(\lambda_m)_{m \geq 0}$ its eigenvalues, taking multiplicity into account, which are the same as those of $T_K$ when data is distributed according to $d\tau$ [8]. We then obtain the following simple result relating $\mathcal{N}_{K_G}$ to $\mathcal{N}_K$.

**Lemma 4** (Degrees of freedom for $K$ and $K_G$.). *For any $\ell \geq 0$, we have*

$$\mathcal{N}_{K_G}(\lambda) \leq D(\ell) + \nu_d(\ell) \mathcal{N}_K(\lambda),$$

*where $D(\ell) := \sum_{k < \ell} \overline{N}(d, k)$ and $\nu_d(\ell) := \sup_{k \geq \ell} \gamma_d(k)$, with $\gamma_d$ given in* (6)*.*

This suggests that for a fixed $\ell$, the effective dimension of $K_G$ is controlled by a factor $\nu_d(\ell)$ times that of $K$, up to a finite fixed dimension $D(\ell)$. For difficult non-parametric problems which require small $\lambda$ at large sample sizes, the second term will tend to dominate, so that having a small $\nu_d(\ell)$ may help reduce sample complexity compared to using the vanilla kernel $K$, an observation which we make rigorous below.

**Generalization bound for KRR.** Armed with the above lemmas on approximation error and degrees of freedom, we now study generalization of KRR under the following assumptions:

(A1) *capacity condition*: $\mathcal{N}_K(\lambda) \leq C_K \lambda^{-1/\alpha}$ with $\alpha > 1$.

(A2) *source condition*: there exists $r > \frac{\alpha-1}{2\alpha}$ and $g \in L^2(d\tau)$ with $\|g\|_{L^2(d\tau)} \leq C_{f^*}$ such that $f^* = T_K^r g$.

(A3) *invariance*: $f^*$ is $G$-invariant.

(A4) *problem noise*: $\rho$ is such that $\mathbb{E}_\rho[(y - f^*(x))^2|x] \leq \sigma_\rho^2$.

The first, second, and fourth conditions are commonly used in the kernel methods literature [8]. (A1) characterizes the "size" of the RKHS, and is satisfied when the eigenvalues $\lambda_m$ of $T_K$ decay as $k^{-\alpha}$. On the sphere, $\alpha = \frac{2s}{d-1}$ corresponds to having $s$ bounded derivatives, *e.g.*, we have $s = d/2 + 1$ for the arc-cosine kernel. The parameter $r$ in (A2) defines the regularity of $f^*$ relative to that of the kernel: $r = 1/2$ corresponds to $f^* \in \mathcal{H}_K$, while larger (resp. smaller) $r$ implies $f^*$ is more (resp. less) smooth. The condition on $r$ is needed for our specific bound, which is based on [3, Proposition 7.2], but may be bypassed using different algorithms or analyses [17, 26]. We now present our bound on the excess risk.

**Theorem 5** (Generalization of invariant kernel.)**.** *Assume (A1-4). Let $\nu_d(\ell)$ be as in Lemma 4, or an upper bound thereof, and assume $\nu_0 := \inf_{\ell \geq 0} \nu_d(\ell) > 0$.*
*Let $n \geq \max\left\{\|f^*\|_\infty^2/\sigma_\rho^2, (C_1/\nu_0)^{\frac{\alpha}{2\alpha r+1-\alpha}}\right\}$, and define*

$$\ell_n := \sup\{\ell : D(\ell) \leq C_2 \nu_d(\ell)^{\frac{2\alpha r}{2\alpha r+1}} n^{\frac{1}{2\alpha r+1}}\}. \tag{13}$$

*We then have, for $\lambda = C_3(\nu_d(\ell_n)/n)^{\alpha/(2\alpha r+1)}$,*

$$\mathbb{E}[R(\hat{f}_\lambda) - R(f^*)] \leq C_4 \left(\frac{\nu_d(\ell_n)}{n}\right)^{\frac{2\alpha r}{2\alpha r+1}}. \tag{14}$$

*In the same setting, KRR with kernel $K$ and $\lambda = C_3 n^{\frac{-\alpha}{2\alpha r+1}}$ achieves $\mathbb{E}[R(\hat{f}_\lambda) - R(f^*)] \leq C_4 n^{\frac{-2\alpha r}{2\alpha r+1}}$, where $C_3, C_4$ are the same constants as for the invariant kernel. Here, the constants $C_{1:4}$ only depend on the parameters of assumptions (A1-4).*

The theorem shows that the generalization error for the invariant kernel behaves as if it effectively had access to $n/\nu_d(\ell_n)$ samples, so that $\nu_d(\ell_n)$ plays the role of an *effective* inverse sample complexity gain at sample size $n$. Note that $\nu_d(\ell_n) \leq 1$ and $\nu_d$ is decreasing, so that we always have some improvement in sample complexity, and this gets better when $\ell_n$ is large. In particular, we show in Section 4 that $\gamma_d(k)$, and hence $\nu_d(\ell)$ converge to $1/|G|$, so that asymptotically the gain in sample complexity can be as large as the size of the group, which in some cases may grow *exponentially* in $d$.

**Asymptotic estimates of the effective gain $\nu_d(\ell_n)$.** We now study the asymptotic behavior of the effective gain factor $\nu_d(\ell_n)$ when $n \to \infty$, by considering a case where an asymptotic equivalent of $\nu_d(\ell)$ in $\ell$ is known:
$$\nu_d(\ell) \approx \nu_0 + c\ell^{-\beta}.$$
In Section 4, we obtain such asymptotics with $\nu_0 = 1/|G|$, and a rate $\beta$ that depends on spectral properties of the elements of $G$, as well as upper bounds with possibly faster rates $\beta$ at the cost of larger $\nu_0$. In Appendix B.5, we show that we may leverage this to obtain the following asymptotic estimate of the effective gain $\nu_d(\ell_n)$:

$$\nu_d(\ell_n) \leq \nu_0 + C \min\left\{(\nu_0^{2\alpha r} n)^{\frac{-\beta}{(d-1)(2\alpha r+1)}}, n^{\frac{-\beta}{(d-1)(2\alpha r+1)+2\beta\alpha r}}\right\}. \tag{15}$$

Notice that when $\beta \ll d$, both exponents of $n$ display a curse of dimensionality, but this curse goes away as $\beta$ grows. Note also that the first exponent yields a faster rate, but one that is only achieved for large $n$ due to the factor $\nu_0^{2\alpha r}$, which may be small for large groups.

**Curse of dimensionality and optimality.** Note that the bound obtained in Theorem 5 is still cursed for an invariant target $f^*$, in the sense that the exponent in the rate is of order $1/d$ when $f^*$ is only assumed to be Lipschitz. Indeed, a Lipschitz assumption on $f^*$ corresponds to taking $r$ and $\alpha$ such that $2\alpha r \approx 2/(d-1)$, which makes the source condition (A2) similar to a bound on $\|\nabla f^*\|_{L^2(d\tau)}$.

This then leads to a cursed rate $n^{-2/(2+d-1)}$, raising the question of whether this can be improved. We note that since $\gamma_d(k) = \Omega(1/|G|)$ (as we show in Section 4), the asymptotic decays (as a function of $k$) of the coefficients of $f^*$ and of the eigenvalues of $T_{K_G}$ are similar to those for the non-invariant case, which implies that these rates cannot be improved (see, *e.g.*, [8]). In Appendix B.6, we show that our bounds with an improvement in sample complexity by a factor $|G|$ are asymptotically minimax optimal, so that this may be the best we can hope for under our assumptions.

**Comparison to [23].** The work [23] also consider non-parametric learning of invariant functions with similar kernels. They consider a high-dimensional regime where $d \to \infty$ with sample sizes in polynomial scalings $n \approx d^s$ for some $s$. They then show that if $\gamma_d(k) = \Theta_d(d^{-\alpha})$ as $d \to \infty$, for some $\alpha > 0$ (which they call *degeneracy*), then the invariant kernel can learn polynomials of degree $\ell$ with $n \approx d^{\ell-\alpha}$ while the non-invariant kernel needs $n \approx d^\ell$ samples. In some cases, such as the cyclic group, [23] show $\alpha = 1$ and hence the gain of a factor $d^\alpha = d$ is equal the size of the group, but in other cases $d^\alpha$ may be smaller than the group size. For groups of size exponential in $d$, the analysis in [23] may only achieve polynomial improvements by factors $d^\alpha$, in contrast to our analysis, which considers the different regime of fixed $d$ and $n \to \infty$, and may lead to gains by exponential factors if $|G|$ is large, at least asymptotically.

# 4 Counting Invariant Polynomials

In this section, we study the asymptotic behavior of $\gamma_d(k)$, given in (6), when $k \to \infty$ and the dimension $d$ and the group $G$ are fixed. This quantity can be seen as capturing the fraction of orthogonal spherical harmonics of degree $k$ that are invariant to $G$, and helps us control the possible gains in sample complexity for learning invariant functions, as described in Section 3. Denoting $\gamma_{d,\sigma}(k) := \mathbb{E}_x[P_{d,k}(\langle \sigma \cdot x, x \rangle)]$, we will show that $\gamma_{d,\sigma}(k)$ vanishes for large $k$ for any $\sigma$ that is not the identity. This implies that $\gamma_d(k)$ converges to $1/|G|$, since we trivially have $\gamma_{d,Id}(k) = P_{d,k}(1) = 1$. We further characterize the asymptotic behavior of $\gamma_d(k)$ in terms of properties of the group elements. In the following we consider the case of $G$ being a subgroup of $S_d$, the groups of permutations on $d$ elements.

**Decay of $\gamma_{d,\sigma}(k)$.** Our main insight is to leverage the fact that when $\sigma$ is not the identity, then the random variable $z_\sigma = \langle \sigma \cdot x, x \rangle$ when $x \sim \tau$ admits a density on $[-1, 1]$, which we denote $q_\sigma$. This by itself will prove sufficient to show that $\gamma_{d,\sigma}(k)$ decays for large $k$, thanks to the oscillatory behavior of $P_{d,k}$. We can then further characterize its asymptotic behavior by studying the singularities of $q_\sigma$, leveraging the seminal work of Saldanha and Tomei [28]. In particular, these depend on spectral properties of the matrix associated to $\sigma$. We summarize this in the next proposition.

**Proposition 6** (Asymptotic behavior of $\gamma_{d,\sigma}(k)$.)**.** *Let $A_\sigma$ be the matrix associated to $\sigma \neq \mathrm{Id}$, that is such that $\sigma \cdot x = A_\sigma x$. Denote by $\Lambda_\sigma$ the set of (complex) eigenvalues of $A_\sigma$, and by $m_\lambda$ the multiplicity of $\lambda \in \Lambda_\sigma$. When $k \to \infty$, we have the asymptotic equivalent $\gamma_{d,\sigma}(k) = \sum_{\lambda \in \Lambda_\sigma} \gamma_{d,\sigma,\lambda}(k)$, where*

$$\gamma_{d,\sigma,\lambda}(k) \lesssim \begin{cases} k^{-d+m_\lambda} + o(k^{-d+m_\lambda}), & \text{if } \lambda \in \{\pm 1\}, \\ k^{-d+m_\lambda+4} + o(k^{-d+m_\lambda+4}), & \text{otherwise,} \end{cases} \tag{16}$$

*where $\lesssim$ hides constants that may depend on $d$, $\sigma$ and $\lambda$.*

Every permutation $\sigma \in \mathcal{S}_d$ (where $\mathcal{S}_d$ is the symmetric group of permutations) can be decomposed into cycles on disjoint orbits; the eigenvalues $\lambda$ (and their multiplicities $m_\lambda$) of a matrix $A_\sigma$ admit an interpretation based on such decomposition. Indeed, since $A_\sigma$ is unitary, its eigenvalues are of the form $\lambda = e^{2\pi i\theta}$, and one can verify that necessarily $\theta = \frac{p}{q} \in \mathbb{Q}$. Furthermore, assuming w.l.o.g. that $q$ is prime, such eigenvalue appears whenever $\sigma$ contains a cycle of length a multiple of $q$. In particular, the multiplicity of the eigenvalue 1, $m_1$, corresponds to the total number of cycles in such a decomposition, which we will denote by $c(\sigma)$. Then $\gamma_{d,\sigma}(k)$ can be controlled as follows.

**Corollary 7** (Decay of $\gamma_{d,\sigma}(k)$)**.** *Let $\sigma \neq \mathrm{Id}$, and let $c(\sigma)$ denote the number of cycles in $\sigma$. Then,*

$$\gamma_{d,\sigma}(k) \lesssim \begin{cases} k^{-d+c(\sigma)}, & \text{if } c(\sigma) > \frac{d+3}{2}, \\ k^{-d/2+6}, & \text{otherwise .} \end{cases}$$

**Decay on specific subgroups.** We may now use Corollary 7 to study the asymptotic behaviour of $\gamma_d(k)$ as $k \to \infty$ for various choices of subgroups of $\mathcal{S}_d$, using the following result.

**Corollary 8** (Upper bounds with permutation statistics). *Let $G$ be a subset of $\mathcal{S}_d$ and define $\zeta(G, s) := |\{\sigma \in G \ : \ c(\sigma) > s\}|$ for $s \in [d-1]$. Then, for any $s$, we have*

$$\gamma_d(k) \leq \frac{\zeta(G, s)}{|G|} + O\left(k^{-d+\max\{s,\, d/2+6\}}\right) , \tag{17}$$

*with equality if $s$ is such that $\zeta(G, s) = 1$.*

Note that such an upper bound immediately yields a similar upper bound for $\nu_d(\ell)$ as defined in Section 3, which then controls the effective gain in sample complexity in Theorem 5. Indeed, (17) implies that there is a constant $C$ such that for all $k > 0$, $\gamma_d(k) \leq \zeta(G, s)/|G| + Ck^{-d+\max\{s,d/2+6\}}$. Since this upper bound decreases with $k$, we obtain

$$\nu_d(\ell) \leq \frac{\zeta(G, s)}{|G|} + C\ell^{-d+\max\{s,d/2+6\}}.$$

In the context of the generalization bound of Theorem 5 and our heuristic derivation thereafter, the *effective* gain in sample complexity is then governed an upper bound on $\nu_d(\ell_n)$ as in (15), with asymptotic gain $\nu_0 = \frac{\zeta(G,s)}{|G|}$ and rate $\beta = d - \max\{s, d/2 + 6\}$.

**Example 9** (Translations). *Let $G = C_d$ be the cyclic group on $d$ elements. Then it holds*

$$\gamma_d(k) = \frac{1}{d} + O\left(k^{-d/2+6}\right) .$$

*This follows by noticing that every translation $\sigma$ (but the identity) satisfies $c(\sigma) \leq d/2$. This leads to an asymptotic gain $\nu_0^{-1} = d$ and $\beta = d/2 - 6$ leads to fast convergence in* (15) *even when $d$ is large.*

**Example 10** (Local translations). *Let $d = s \cdot r$ (with $r, s \geq 5$ for simplicity), and consider the group composed of traslations over $r$ blocks of coordinates of size $s$; i.e., the block-cyclic group*

$$G = \{\sigma \ : \ \sigma = \sigma^{(1)} \circ \cdots \circ \sigma^{(r)}\}$$

*where each $\sigma^{(i)}$ is a translation over the set $\{(i-1)s+1, \ldots, is\}$, for $i \in [r]$. Then it holds*

$$\gamma_d(k) = \frac{1}{s^r} + O\left(k^{-s/2+1}\right) . \tag{18}$$

*This follows by noticing that every local translation $\sigma$ (but the identity) satisfies $c(\sigma) \leq (d-s) + s/2$. Here the asymptotic gain is $\nu_0^{-1} = s^r = s^{d/s}$, which can be exponential in $d$ when $s$ is small. We have $\beta = s/2 - 1$, which leads to much slower convergence than the translation case, unless $s$ is large and of order $d$.*

**Example 11** (Full permutation group). *For the case of $G = \mathcal{S}_d$, we can split the group based on the value of $\mathrm{Fix}(\sigma)$, the number of elements fixed by a permutation $\sigma$. Denote*

$$\xi(G, s) := |\sigma \in G \ : \ \mathrm{Fix}(\sigma) > s| = \sum_{j=s+1}^{d} \binom{d}{j} !(d-j) .$$

*for $s \in [d-1]$, where $!k$ denotes the $k$-th subfactorial. Then we have*

$$\gamma_d(k) \leq \frac{\xi(G, s)}{d!} + O\left(k^{-d/2+\max\{s/2,\, 6\}}\right) ,$$

*with equality for $s = d - 1$. This follows from the fact that $c(\sigma) \leq \mathrm{Fix}(\sigma) + (d - \mathrm{Fix}(\sigma))/2$. In particular, it follows*

$$\gamma_d(k) \leq \frac{2}{(s+1)!} + O\left(k^{-d/2+\max\{s/2,\, 6\}}\right).$$

*When considering the full group, we may get a large asymptotic improvement of order $\nu_0^{-1} = |G| = d!$ in sample complexity, but a slow convergence with $\beta = -1$ as per Corollary 8 (assuming $d$ large enough). Using different values of $s$ may yield different upper bounds with faster convergence rates $\beta = d/2 - \max\{s/2, 6\}$, but smaller asymptotic gains in sample complexity, given by $\nu_0^{-1} = (s+1)!/2$. For instance, with $s = d/2$ and $d$ large enough, we have $\beta = d/4$, leading to a potentially fast convergence rate in 15 towards a sample complexity gain that is still significantly large, of order $(d/2 + 1)!/2$.*

Overall, these examples show that the size of the group determines the best possible improvement in sample complexity, while the spectral properties of its permutations dictate how quickly we may achieve these gains.

# 5 Beyond Group Invariance: Geometric Stability

In this section, we study gains in sample complexity when the target function $f^*$ is not fully invariant to a group, but may be stable under small geometric changes on the input. We formalize this by considering a similar averaging operator $S_G$, but we allow $G$ to be a generic set of permutations instead of a group, and allow for a weighted average:

$$S_G f(x) := \sum_{\sigma \in G} h(\sigma) f(\sigma \cdot x), \tag{19}$$

where $h(\sigma) \geq 0$ for all $\sigma \in G$ and $\sum_{\sigma \in G} h(\sigma) = 1$. We assume that $G$ is "symmetric", *i.e.*, $\sigma^{-1} \in G$ when $\sigma \in G$ and $h(\sigma^{-1}) = h(\sigma)$, so that $S_G$ is self-adjoint. In this case, images of $S_G$ are not invariant functions, but may nevertheless exhibit a form of "local" stability to small perturbations of the input data. For instance, if $G$ consists of local translations by at most a few pixels, or if $G$ consists of all translations but $h$ is localized around the identity, then applying $S_G$ yields functions that are stable to local translations. We may also consider a more structured set $G$ of permutations that resemble local deformations, consisting of both a global translation as well as different local translations at different scales, as we describe below.

**Spectral properties of $S_G$.** Note that in this setup, we no longer have that $S_G$ is a projection, however we may still view it as a *smoothing* operator, which attenuates certain harmonics that are "less" invariant than others. The next lemma shows related spectral properties to the invariant case.

**Lemma 12** (Spectral properties of $S_G$). *There exists a basis of spherical harmonics $\overline{Y}_{k,j}$, for $k \geq 0$, and $j = 1, \ldots, N(d, k)$, in which the operator $S_G$ is diagonal, with eigenvalues $\lambda_{k,j} \geq 0$. In analogy to Lemma 1, we have*

$$\gamma_d(k) := N(d, k)^{-1} \sum_{j=1}^{N(d,k)} \lambda_{k,j} = \sum_{\sigma \in G} h(\sigma) \mathbb{E}_x \left[ P_{d,k}(\langle \sigma \cdot x, x \rangle) \right]. \tag{20}$$

*We also define $\nu_d(\ell) := \sup_{k \geq \ell} \gamma_d(k)$.*

**Sample complexity of stable kernel.** In analogy to Section 3, we may consider a stable kernel

$$K_G(x, x') = \sum_{\sigma \in G} h(\sigma) \kappa(\langle \sigma \cdot x, x' \rangle). \tag{21}$$

Then, it is easy to check that the integral operator of $K_G$ is given by $T_{K_G} = S_G T_K$. In contrast to Section 3, we no longer have that the approximation errors of $K$ and $K_G$ are the same in general on "geometrically stable" functions, since the notion is not precisely defined. Nevertheless, we may represent favorable targets $f^*$ as those whose coefficients decay similarly at each frequency $k$ to those of $S_G$, by viewing it as a smoothing of some $L^2$ function $g^*$, *i.e.*, $f^* = S_G^r g^*$ for some exponent $r$. With this in mind, we make the following assumptions, replacing assumptions (A1-3) of Section 3.

(A5) *capacity*: the eigenvalues $(\xi_m)_{m \geq 0}$ of $T_K$ satisfy $\xi_m \leq C(m + 1)^{-\alpha}$.

(A6) *source condition*: there exists $r > \frac{\alpha - 1}{2\alpha}$ and $g \in L^2(d\tau)$ with $\|g\|_{L^2(d\tau)} \leq C_{f^*}$ such that $f^* = S_G^r T_K^r g$.

Note that (A6) corresponds to a standard source condition with the kernel $K_G$ (since $T_{K_G}^r = S_G^r T_K^r$), yet it reveals how $K_G$ jointly performs smoothing on the sphere, through $T_K$, as well as on permutations through $S_G$. While these two forms of smoothing appear "entangled" in this assumption, one may balance them by choosing different levels of smoothing in the kernel function $\kappa$, or by averaging multiple times in (21). Assumption (A5) is needed for obtaining a variant of Lemma 4, and implies (A1) with $C_K \propto C^{1/\alpha}$. We then obtain the following generalization bound.

**Theorem 13** (Generalization with geometric stability.). *Assume (A4-6), and assume $\nu_0 := \inf_{\ell \geq 0} \nu_d(\ell) > 0$. Let $n \geq \max(\|f^*\|_\infty^2 / \sigma_\rho^2, (C_1/\nu_0)^{1/(2\alpha r + 1 - \alpha)})$, and define*

$$\ell_n := \sup\{\ell : D(\ell) \leq C_2 \nu_d(\ell)^{\frac{2r}{2\alpha r + 1}} n^{\frac{1}{2\alpha r + 1}}\}. \tag{22}$$

*We then have, for $\lambda = C_3(\nu_d(\ell_n)^{1/\alpha}/n)^{\alpha/(2\alpha r+1)}$,*

$$\mathbb{E}[R(\hat{f}_\lambda) - R(f^*)] \leq C_4 \left( \frac{\nu_d(\ell_n)^{1/\alpha}}{n} \right)^{\frac{2\alpha r}{2\alpha r+1}}. \tag{23}$$

*In the same setting, KRR with kernel $K$ achieves a similar bound with $\nu_d(\ell_n)^{1/\alpha}$ replaced by 1, but with a possibly smaller constant $C_4$. Here, the constants $C_{1:4}$ only depend on the parameters of assumptions (A4-6).*

Note that the obtained generalization bound is very similar to Theorem 5, but with a factor $\nu_d(\ell_n)^{1/\alpha}$ instead of $\nu_d(\ell_n)$. This is due to the fact that in contrast to the invariant case, where $\gamma_d(k)$ in (6) can help precisely control the number of invariant spherical harmonics, in this case $\gamma_d(k)$ as computed in (20) can only give information about the sum of the eigenvalues $\lambda_{k,j}$ at frequency $k$, which may be insufficient to precisely estimate the gains in effective dimension. The gap between these two factors is relatively small for kernels with slow decays ($\alpha \approx 1$) but can be more pronounced for smooth kernels with fast decays (large $\alpha$). Note also that the different source condition (A6) leads to a different approximation error, and thus to an approximation-estimation trade-off related to stability, which does not appear in the group-invariant case. More precise estimates of the decays of $\lambda_{k,j}$ may help characterize this tradeoff more formally, and we leave this question to future work. As in (15), we may derive an estimate of $\nu_d(\ell_n)$, namely if $\nu_d(\ell) \approx \nu_0 + c\ell^{-\beta}$, then we have

$$\nu_d(\ell_n) \leq \nu_0 + C \min \left\{ (\nu_0^{2r} n)^{\frac{-\beta}{(d-1)(2\alpha r+1)}}, n^{\frac{-\beta}{(d-1)(2\alpha r+1)+2\beta r}} \right\}. \tag{24}$$

**Deformation-like stability.**  For inputs $x$ defined as signals $x \in L^2(\Omega)$ over a continuous domain $\Omega \subseteq \mathbb{R}^s$, $s = 1, 2$, the action of 'small' diffeomorphisms $\varphi : \Omega \to \Omega$ as $(\varphi \cdot x)(u) = x(\varphi^{-1}(u))$ is a powerful diagnostic of performance of trainable CNNs [25], and a key guiding principle for scattering representations [22, 7]. In these works, the basic deformation cost is measured as $\|\varphi\| := \sup_u \|\nabla \varphi(u) - \mathbf{I}\|$. We instantiate an equivalent of small deformations in our finite-dimensional setting as follows.

$$\Phi_\varepsilon := \{ \sigma \in \mathcal{S}_d \ : \ |\sigma(u) - \sigma(u') - (u - u')| \leq \varepsilon |u - u'| \}, \tag{25}$$

where the differences are taken modulo $d$. For $\varepsilon = 0$, we recover the translation group described in Example 9. We can verify that $\varepsilon = 1$ also corresponds to the translation group (due to the constraint that $\sigma(u) \neq \sigma(u')$ whenever $u \neq u'$), thus the first non-trivial model corresponds to $\varepsilon = 2$.

**Proposition 14** (Upper bound on $\gamma_d(k)$ for deformations.). *It holds $\Phi_2^{-1} = \Phi_2$. Moreover, $|\Phi_2| \geq \tau^d$ for $\tau \approx 1.714$, and*

$$\gamma_d(k) \leq C \left( \frac{e^{2\eta}}{(2\eta)^{2\eta}\tau^{1-2\eta}} \right)^d + O\left( k^{-\eta d} \right) \tag{26}$$

*for $\eta < 1/4$. In particular, $\tilde{\tau}^{-1} := \frac{e^{2\eta}}{(2\eta)^{2\eta}\tau^{1-2\eta}} < 1$ for $\eta < 0.07$, leading to an effective gain in sample complexity exponential in $d$, $\nu_0^{-1/\alpha} = \Theta(\tilde{\tau}^{d/\alpha})$; and $\beta = \eta d$ resulting in fast convergence in (24) even for large $d$.*

We thus verify that small deformations, already with $\varepsilon = 2$, provide a substantial gain relative to rigid translations, since $\Phi_2$ now grows exponentially with the dimension, rather than linearly. Let us remark that our small deformation model (25) acting on $\{1, d\}$ differs in important ways from diffeomorphisms acting on a continuous domain. In our case they define unitary operators (since they are constructed as subsets of the permutation group), as opposed to diffeorphims, for which $\|\varphi \cdot x\|_{L^2(\Omega)} \neq \|x\|_{L^2(\Omega)}$ generally. In other words, the 'deformations' in $\Phi_\varepsilon$ are more akin to local shufflings of the pixels rather than local distortions. That said, our model does roughly capture the size of small deformation classes. An interesting question for future work is to extend our framework to non-unitary transformations, which could accommodate appropriate discretisations of continuous diffeomorphisms.

## 6   Numerical Experiments

In this section, we provide simple numerical experiments on synthetic data which illustrate our theoretical results. In Figure 1, we consider KRR on 5 000 training samples with inputs uniformly

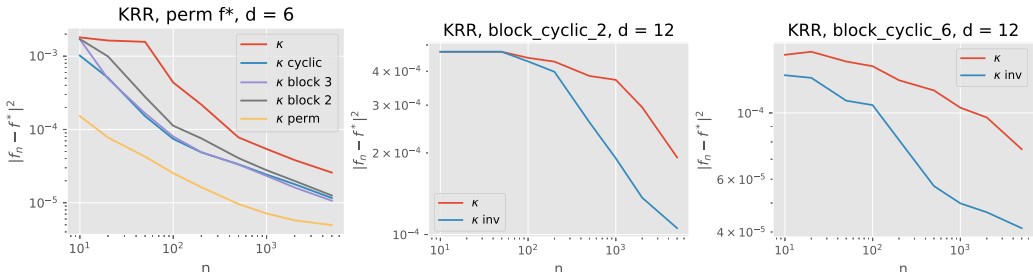

Figure 1: Comparison of KRR with invariant and non-invariant kernels. (left) permutation-invariant target with $d = 6$, comparison between various invariant kernels (cyclic, block-cyclic, and permutation groups). (center/right) invariant vs non-invariant kernels on invariant target functions with $d = 12$, for block-cyclic groups $G$ of two different sizes.

distributed on $\mathbb{S}^{d-1}$, and outputs generated according to a target non-smooth function $f^* = S_G g^*$, with $g^*(x) = \mathbb{1}\{w_*^\top x \geq 0.7\}$, where the averaging operator $S_G$ is over different groups in each plot. The regularization parameter $\lambda$ is optimized on $5\,000$ test samples. We use the dot-product kernel function $\kappa(u) = (u + 1)\kappa_1(u)$, where $\kappa_1$ is the arc-cosine kernel of degree 1, which corresponds to an infinite-width shallow ReLU network [9].

When the target is permutation-invariant, we can see in Figure 1(left) that the kernel based on permutation invariance leads to the largest gain in sample complexity compared to those which use cyclic or block-cyclic groups. Since the permutation group has the largest cardinality, this is consistent with our finding that the gains may be of the order of the size of the group. Figures 1(center/right) consider the example of cyclic translations on local blocks of size 2 or 6 (Example 10 with $s = 2$ or 6), and illustrate that the improvement in sample complexity happens later for $s = 2$ than $s = 6$, which is consistent with the slower decays of $\gamma_d$ obtained in (18) due to the larger number of cycles.

## 7   Discussion and Conclusion

We have studied how geometric invariance or stability assumptions on target functions enable more efficient learning, with improvements in sample complexity which may be as large as the number of permutations considered in the group or set of elements to which the target is invariant or stable. In particular, this gain can be exponential in the dimension if we consider, *e.g.*, all permutations, local translations on small blocks, or permutations that resemble small deformations. This last example provides a strong justification for seeking models and architectures that are stable to deformations, a natural prior when learning functions on images [25]. In that respect, our results provide a theoretical baseline to assess learning guarantees under geometric priors: by designing appropriate geometrically stable kernels, we simultaneously address approximation and generalisation errors within a framework of convex optimization.

That said, while these gains may be large in practice, the obtained rates are still generically cursed by dimension if the target is non-smooth. In other words, invariance or geoemtric stability allows us to express Lipschitz assumptions with respect to weaker metrics. While these stronger regularity assumptions result in important gains in sample complexity, they do not overcome the inherent difficulty of learning non-smooth structures in high-dimensions. This suggests that further assumptions may be needed to learn efficiently on high-dimensional geometric data, for instance with more structured forms of regularity beyond our invariant/stable setup, which may be exploited perhaps through architectures that involve local connectivity, hierarchy (which would explain the benefits of depth, as opposed to our current results), or feature learning [2, 4, 16, 21, 27]. A natural further question is to study stability to transformations that are not necessarily permutations, and which may then provide more realistic models of continuous deformations. Another interesting question is to study whether it is possible to adapt to general symmetries present in the target, instead of encoding them in the model with an appropriately designed kernel as done here.

## Acknowledgments and Disclosure of Funding

LV and JB acknowledge partial support from the Alfred P. Sloan Foundation, NSF RI-1816753, NSF CAREER CIF 1845360, NSF CHS-1901091, and Samsung Electronics.

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
