This appendix contains additional background on spherical harmonics (Appendix A), and proofs of the results from Section 3, 4 and 5 (in Appendix B, C and D, respectively).

## A  Background on Spherical Harmonics and Legendre Polynomials

In this section, we provide background on spherical harmonic and Legendre/Gegenbauer polynomials, which are used extensively in our analysis. See, *e.g.*, [1, 14] for references. We consider inputs on the $d-1$ sphere $\mathbb{S}^{d-1} = \{x \in \mathbb{R}^d, \|x\| = 1\}$.

We recall some properties of the spherical harmonics $Y_{k,j}$ introduced in Section 2. For $j = 1, \ldots, N(d,k)$, where $N(d,k) = \frac{2k+d-2}{k}\binom{k+d-3}{d-2}$, the spherical harmonics $Y_{k,j}$ are homogeneous harmonic polynomials of degree $k$ that are orthonormal with respect to the uniform distribution $\tau$ on the $d-1$ sphere. The degree $k$ plays the role of an integer frequency, as in Fourier series, and the collection $\{Y_{k,j}, k \geq 0, j = 1, \ldots, N(d,k)\}$ forms an orthonormal basis of $L^2(\mathbb{S}^{d-1}, d\tau)$. As with Fourier series, there are tight connections between decay of coefficients in this basis w.r.t. $k$, and regularity/differentiability of functions, in this case differentiability on the sphere. In particular, this is a key property that we exploit for obtaining decays related to the number of invariant polynomials in Proposition 6. This follows from the fact that spherical harmonics are eigenfunctions of the Laplace-Beltrami operator on the sphere $\Delta_{\mathbb{S}^{d-1}}$ (see [14, Proposition 4.5]):

$$\Delta_{\mathbb{S}^{d-1}} Y_{k,j} = -k(k+d-2)Y_{k,j}. \tag{27}$$

For a given frequency $k$, we have the following addition formula:

$$\sum_{j=1}^{N(d,k)} Y_{k,j}(x)Y_{k,j}(y) = N(d,k)P_{d,k}(x^\top y), \tag{28}$$

where $P_{d,k}$ is the $k$-th Legendre polynomial in dimension $d$ (also known as Gegenbauer polynomial when using a different scaling), given by the Rodrigues formula:

$$P_{d,k}(t) = (-1/2)^k \frac{\Gamma(\frac{d-1}{2})}{\Gamma(k+\frac{d-1}{2})}(1-t^2)^{(3-d)/2}\left(\frac{d}{dt}\right)^k (1-t^2)^{k+(d-3)/2}. \tag{29}$$

The polynomials $P_{d,k}$ are orthogonal in $L^2([-1,1], dw)$ where the measure $dw$ is given by the weight function $dw(t) = (1-t^2)^{(d-3)/2}dt$, and we have

$$\int_{-1}^{1} P_{d,k}^2(t)(1-t^2)^{(d-3)/2}dt = \frac{\omega_{d-1}}{\omega_{d-2}}\frac{1}{N(d,k)}, \tag{30}$$

where $\omega_{p-1} = \frac{2\pi^{p/2}}{\Gamma(p/2)}$ denotes the surface of the sphere $\mathbb{S}^{p-1}$ in $p$ dimensions. Using the addition formula (28) and orthogonality of spherical harmonics, we can show

$$\int P_{d,j}(w^\top x)P_{d,k}(w^\top y)d\tau(w) = \frac{\delta_{jk}}{N(d,k)}P_{d,k}(x^\top y) \tag{31}$$

We will use the following recurrence relation of Legendre polynomials [14, Eq. 4.36]

$$tP_{d,k}(t) = \frac{k}{2k+d-2}P_{d,k-1}(t) + \frac{k+d-2}{2k+d-2}P_{d,k+1}(t), \tag{32}$$

for $k \geq 1$, and for $k = 0$ we simply have $tP_{d,0}(t) = P_{d,1}(t)$. We will also use the following pointwise upper bound on $P_{d,k}(t)$ (see [1, Eq. 2.117]):

$$|P_{d,k}(t)| \leq \frac{\Gamma\left(\frac{d-1}{2}\right)}{\sqrt{\pi}}\left(\frac{4}{k(1-t^2)}\right)^{(d-2)/2} \tag{33}$$

The Funk-Hecke formula is helpful for computing Fourier coefficients in the basis of spherical harmonics in terms of Legendre polynomials: for any $j = 1, \ldots, N(d,k)$, we have

$$\int f(x^\top y)Y_{k,j}(y)d\tau(y) = \frac{\omega_{d-2}}{\omega_{d-1}}Y_{k,j}(x)\int_{-1}^{1} f(t)P_{d,k}(t)(1-t^2)^{(d-3)/2}dt. \tag{34}$$

For example, we may use this to obtain decompositions of dot-product kernels by computing Fourier coefficients of functions $\kappa(\langle x, \cdot \rangle)$. Indeed, denoting

$$\mu_k = \frac{\omega_{d-2}}{\omega_{d-1}} \int_{-1}^{1} \kappa(t) P_{d,k}(t)(1-t^2)^{(d-3)/2} dt,$$

writing the decomposition of $\kappa(\langle x, \cdot \rangle)$ using (34) leads to the following Mercer decomposition of the kernel:

$$\kappa(x^\top y) = \sum_{k=0}^{\infty} \mu_k \sum_{j=1}^{N(d,k)} Y_{k,j}(x) Y_{k,j}(y) = \sum_{k=0}^{\infty} \mu_k N(d,k) P_{d,k}(x^\top y). \tag{35}$$

# B  Proofs for Section 3 (Sample complexity of Invariant Kernels)

## B.1  Proof of Lemma 2 (spectral properties of $K$ and $K_G$)

*Proof.* That $\mu_k$ are eigenvalues of $T_K$ with multiplicity $N(d,k)$ is standard and follows from the Funk-Hecke formula (see, *e.g.*, [29]). In particular, for any spherical harmonic $Y_k \in V_{d,k}$, we have $T_K Y_k = \mu_k Y_k$, and there are $N(d,k)$ orthogonal spherical harmonics in $V_{d,k}$.

For $K_G$, note that we have $T_{K_G} = S_G T_K$, so that for any $G$-invariant spherical harmonic $\overline{Y}_k \in \overline{V}_{d,k}$ we have $T_{K_G} \overline{Y}_k = \mu_k S_G \overline{Y}_k = \mu_k \overline{Y}_k$, while for any $Y_k \in V_{d,k} \cap \overline{V}_{d,k}^\perp$, we have $T_{K_G} Y_k = 0$ since $S_G Y_k = 0$. $\square$

## B.2  Proof of Lemma 3 (approximation error)

*Proof.* Let $\overline{Y}_{k,j}$, for $j = 1, \ldots, N(d,k)$ be an orthonormal basis of $V_{d,k}$ such that $(\overline{Y}_{k,j})_{j \leq \overline{N}(d,k)}$ form an orthonormal basis of $\overline{V}_{d,k}$. Then, the collection of $\overline{Y}_{k,j}$ for $k \geq 0$ and $j = 1, \ldots, N(d,k)$ forms an orthonormal basis of $L^2(d\tau)$.

For a function $f \in L^2(d\tau)$ with decomposition

$$f(x) = \sum_{k \geq 0} \sum_{j=1}^{N(d,k)} a_{k,j} \overline{Y}_{k,j}(x),$$

we have the following expressions of the RKHS norms for $K$ and $K_G$ by Mercer's theorem (*e.g.*, [12]):

$$\|f\|_{\mathcal{H}_K}^2 = \begin{cases} \sum_{k:\mu_k>0} \sum_{j=1}^{N(d,k)} \frac{a_{k,j}^2}{\mu_k}, & \text{if } a_{k,j} = 0 \text{ whenever } \mu_k = 0 \\ \infty, & \text{otherwise.} \end{cases}$$

$$\|f\|_{\mathcal{H}_{K_G}}^2 = \begin{cases} \sum_{k:\mu_k>0} \sum_{j=1}^{\overline{N}(d,k)} \frac{a_{k,j}^2}{\mu_k}, & \text{if } a_{k,j} = 0 \text{ whenever } \mu_k = 0 \text{ or } j > \overline{N}(d,k) \\ \infty, & \text{otherwise.} \end{cases}$$

Assume now that $f^*$ is invariant, so that its coefficients $a_{k,j}^*$ satisfy $a_{k,j}^* = 0$ for $j > \overline{N}(d,k)$. We have

$$A_{\mathcal{H}_K}(\lambda, f^*) = \inf_{f \in \mathcal{H}_K} \|f - f^*\|_{L^2(d\tau)}^2 + \lambda \|f\|_{\mathcal{H}_K}^2$$

$$= \inf_{a_{k,j}} \sum_{k:\mu_k>0} \sum_{j=0}^{N(d,k)} \left( (a_{k,j} - a_{k,j}^*)^2 + \lambda \frac{a_{k,j}^2}{\mu_k} \right)$$

$$= \inf_{a_{k,j}} \sum_{k:\mu_k>0} \sum_{j=0}^{\overline{N}(d,k)} \left( (a_{k,j} - a_{k,j}^*)^2 + \lambda \frac{a_{k,j}^2}{\mu_k} \right) + \sum_{k:\mu_k>0} \sum_{j=\overline{N}(d,k)+1}^{N(d,k)} (1 + \lambda/\mu_k) a_{k,j}^2$$

$$= \inf_{a_{k,j}} \sum_{k:\mu_k>0} \sum_{j=0}^{\overline{N}(d,k)} \left( (a_{k,j} - a_{k,j}^*)^2 + \lambda \frac{a_{k,j}^2}{\mu_k} \right)$$

$$= A_{\mathcal{H}_{K_G}}(\lambda, f^*),$$

which proves the lemma. $\square$

## B.3 Proof of Lemma 4 (degrees of freedom)

*Proof.* The result immediately follows from the following expressions of degrees of freedom for $K$ and $K_G$:

$$\mathcal{N}_K(\lambda) = \sum_{k \geq 0} N(d, k) \frac{\mu_k}{\mu_k + \lambda}, \qquad \mathcal{N}_{K_G}(\lambda) = \sum_{k \geq 0} \overline{N}(d, k) \frac{\mu_k}{\mu_k + \lambda}. \tag{36}$$

□

## B.4 Proof of Theorem 5 (generalization bound)

*Proof.* We start from the following bound from [3, Proposition 7.2], which holds for any $\lambda \leq 1$, assuming $K_G(x, x) \leq 1$ almost surely (this is satisfied when $\kappa(u) \leq 1$), and for $n \geq \frac{5}{\lambda}(1 + \log(1/\lambda))$:

$$\mathbb{E}[R(\hat{f}_\lambda)] - R(f^*) \leq 16 \frac{\sigma_q^2}{n} \mathcal{N}_{K_G}(\lambda) + 16 A_{\mathcal{H}_{K_G}}(\lambda, f^*) + \frac{24}{n^2} \|f^*\|_\infty^2. \tag{37}$$

Under assumption (A2), we have (see, *e.g.*, [12, Theorem 3, p.33], using that $\|f\|_{\mathcal{H}_K} = \|T_K^{-1/2} f\|_{L^2(d\tau)}$)

$$A_{\mathcal{H}_K}(\lambda, f^*) \leq C_{f^*}^2 \lambda^{2r}, \tag{38}$$

with $C_{f^*} := \|T_K^{-r} f^*\|_{L^2(d\tau)}$ By Lemma 3, we also have

$$A_{\mathcal{H}_{K_G}}(\lambda, f^*) \leq C_{f^*}^2 \lambda^{2r}. \tag{39}$$

Using Lemma 4 for some integer $\ell \geq 0$ and (A1), the bound (37) becomes

$$\mathbb{E}[R(\hat{f}_\lambda)] - R(f^*) \leq 16 C_{f^*}^2 \lambda^{2r} + 16 \frac{\sigma_q^2 D(\ell)}{n} + 16 \frac{C_K \sigma_q^2 \nu_d(\ell)}{n} \lambda^{-1/\alpha} + \frac{24}{n^2} \|f^*\|_\infty^2. \tag{40}$$

Jointly optimizing the first and third terms for $\lambda$ yields

$$\lambda_n = \left( \frac{C_K \sigma_q^2 \nu_d(\ell)}{2r\alpha C_{f^*}^2 n} \right)^{\frac{\alpha}{2\alpha r + 1}} \tag{41}$$

The bound then becomes

$$\mathbb{E}[R(\hat{f}_{\lambda_n})] - R(f^*) \lesssim C_{f^*}^{\frac{2}{2\alpha r + 1}} \left( \frac{C_K \sigma_q^2 \nu_d(\ell)}{n} \right)^{\frac{2\alpha r}{2\alpha r + 1}} + \frac{\sigma_q^2 D(\ell)}{n} + \frac{1}{n^2} \|f^*\|_\infty^2. \tag{42}$$

Here, $\lesssim$ hides only absolute constants that depend on $\alpha$ and $r$. Now we choose $\ell = \ell_n$, given by (13), with constant corresponding to:

$$\ell_n := \sup\{\ell : \sigma_q^2 D(\ell) \leq C_{f^*}^{\frac{2}{2\alpha r + 1}} \left( C_K \sigma_q^2 \nu_d(\ell) \right)^{\frac{2\alpha r}{2\alpha r + 1}} n^{\frac{1}{2\alpha r + 1}} \}, \tag{43}$$

so that the second term is smaller than the first term. The last term is of the same order when

$$n \gtrsim \frac{\|f^*\|_\infty^2}{\sigma_q^2 D(\ell_n)}, \tag{44}$$

which is verified under the condition

$$n \geq \max \left\{ \|f^*\|_\infty^2 / \sigma_\rho^2, (C_1/\nu_0)^{\frac{\alpha}{2\alpha r + 1 - \alpha}} \right\} \tag{45}$$

from the theorem statement, since $D(\ell_n) \geq 1$. Note that for the specific bound (37) to hold we also need $\lambda_n \gtrsim 1/n$ (up to logarithmic terms). This imposes the qualification condition $r > (\alpha - 1)/2\alpha$, and leads to the additional requirement

$$n \gtrsim \left( \frac{C_{f^*}^2}{\sigma_q^2 C_K \nu_d(\ell)} \right)^{\frac{\alpha}{2\alpha r + 1 - \alpha}}. \tag{46}$$

This is verified under condition (45) with $C_1 = C_{f^*}^2 / \sigma_q^2 C_K$.

For the KRR estimator with kernel $K$, the same bound (40) holds, but without the factor $\nu_d(\ell)$ and without the term involving $D(\ell)$. The resulting bound follows from a similar analysis. □

## B.5 Estimating $\ell_n$ and the effective gain $\nu_d(\ell_n)$

In this section, we provide more details on our study of the asymptotic behavior of the quantities $\ell_n$ and $\nu_d(\ell_n)$ in Theorem 5, as described in Section 3.

Since $D(\ell)$ increases with $\ell$, Eq. (13) suggests that $\ell_n$ increases with $n$. We now provide intuition on how we might expect $\ell_n$ and $\nu_d(\ell_n)$ to behave in a situation of interest where we know an asymptotic equivalent of $\nu_d$. Namely, assume that

$$\nu_d(\ell) \approx \nu_0 + c\ell^{-\beta}.$$

We provide such asymptotic equivalents in Section 4, where $\nu_0 = 1/|G|$, and $\beta$ depends on spectral properties of the elements of $G$. For some large groups, $\beta$ may be small, in which case we may consider other approximations with larger $\beta$, at the cost of a larger $\nu_0$. When $\ell_n$ is large, using the approximation $N(d,k) \approx k^{d-2}$, we have $D(\ell) \approx \sum_{k=0}^{\ell-1} k^{d-2} \approx \ell^{d-1}$. Hiding constants other than $\nu_0$, we may then consider $\ell_n$ to be solution of

$$\ell^{\frac{(d-1)(2\alpha r+1)}{2\alpha r}} = n^{\frac{1}{2\alpha r}}(\nu_0 + \ell^{-\beta}).$$

Since the l.h.s. increases, while the r.h.s. decreases with $\ell$, we must have $\ell_n \geq \max(\ell_{n,1}, \ell_{n,2})$, with

$$\ell_{n,1}^{\frac{(d-1)(2\alpha r+1)}{2\alpha r}} = n^{\frac{1}{2\alpha r}}\nu_0, \quad \text{and} \quad \ell_{n,2}^{\frac{(d-1)(2\alpha r+1)}{2\alpha r}} = n^{\frac{1}{2\alpha r}}\ell_{n,2}^{-\beta}.$$

This yields

$$\nu_d(\ell_n) \leq \nu_0 + C\min\left\{(\nu_0^{2\alpha r}n)^{\frac{-\beta}{(d-1)(2\alpha r+1)}}, n^{\frac{-\beta}{(d-1)(2\alpha r+1)+2\beta\alpha r}}\right\}. \tag{47}$$

Notice that when $\beta \ll d$, both exponents of $n$ display a curse of dimensionality, but this curse goes away as $\beta$ grows. Note also that the first exponent yields a faster rate, but one that is only achieved for large $n$ due to the factor $\nu_0^{2\alpha r}$, which may be small for large groups.

## B.6 Discussion of Optimality

In this section, we discuss the optimality of the upper bounds in Theorem 5, in particular the constant $C_4$ and its dependence on the constants $C_K$ and $C_{f^*}$ from the source and capacity conditions.

**Tightness of $C_4$ for non-invariant targets.** We first provide a minimax lower bound for the class of (non-invariant) targets satisfying the source and capacity conditions (A1/A2), in order to show that the constant $C_4$ can be tight (up to absolute constants) in a minimax sense for this class.

Consider a kernel $K_0(x, x') = \kappa_0(\langle x, x'\rangle)$ such that we have the following asymptotics on the eigenvalues of the integral operator: $\lambda_m(T_{K_0}) \sim C_0 m^{-\alpha}$. Let $\ell_0$ be such that for all $m \geq M_0 := D(\ell_0) + 1$ we have $C_0 m^{-\alpha}/2 \leq \lambda_m(T_{K_0}) \leq 2C_0 m^{-\alpha}$. We can then construct a function $\kappa$ and corresponding kernel $K$ such that $\lambda_m(T_K) \leq 2C_0 m^{-\alpha}$ and for $m \geq M_0$, $\lambda_m(T_K) \geq C_0 m^{-\alpha}/2$. For instance, we may define $\kappa(u) = \sum_{k\geq 0} \mu_k(\kappa)N(d,k)P_{d,k}(u)$, with

$$\mu_k(\kappa) = \begin{cases} 2C_0 M_0^{-\alpha} & \text{if } k \leq \ell_0 \\ \mu_k(\kappa_0) & \text{otherwise,} \end{cases}$$

where the $\mu_k(\kappa_0)$ are the Legendre coefficients of $\kappa_0$.

Note that this kernel $K$ satisfies the capacity condition (A1) with $C_K \lesssim C_0^{1/\alpha}$. With this choice of $K$, define $\mathcal{F}$ to be the set of regression functions $f^*$ that further satisfy assumption (A2) with parameters $C_*$ and $r$, and assume that labels are generated as $y = f^*(x) + \epsilon$, with $\epsilon \sim \mathcal{N}(0, \sigma_\rho^2)$. Under these assumptions, note that the upper bound in Theorem 5 is given by

$$\mathbb{E}[\|\hat{f} - f^*\|^2] \lesssim C_*^{\frac{2}{2\alpha r+1}} C_0^{\frac{2r}{2\alpha r+1}}\left(\frac{\sigma_\rho^2}{n}\right)^{\frac{2\alpha r}{2\alpha r+1}}, \tag{48}$$

where $\lesssim$ hides absolute constants or constants depending only on $\alpha$ and $r$.

Following [3], we use Fano's inequality to lower bound the minimax risk. In particular, we have a lower bound

$$M_n(\mathcal{F}) := \inf_{\hat{f}} \sup_{f^*\in\mathcal{F}} \mathbb{E}_{\mathcal{D}_n\sim\rho^{\otimes n}}[\|\hat{f}_{\mathcal{D}_n} - f^*\|_{L^2(d\tau)}^2] \geq A/2,$$

on the minimax risk $M_n(\mathcal{F})$ if we can find a set $\{f_1, \ldots, f_M\} \in \mathcal{F}$, $M \geq 16$, such that

- $\|f_i - f_j\|^2_{L^2(d\tau)} \geq 4A$ for $i \neq j$ (*i.e.*, we have a packing set)
- $\frac{n}{2\sigma_\rho^2}\|f_i - f_j\|^2_{L^2(d\tau)} \leq \frac{\log M}{4}$ (this ensures $f_i$ and $f_j$ are difficult enough to distinguish).

In order to construct a packing, we use the Varshamov-Gilbert lemma to obtain $M \geq \exp(K/8)$ elements $x_1, \ldots, x_M \in \{0,1\}^K$ for some $K$ to be chosen later, which satisfy $\|x_i - x_j\|_1 \geq K/4$ for $i \neq j$. Defining $f_i = \beta \sum_{m=1}^K 2((x_i)_m - 1)\phi_m$, where $(\phi_m)_m$ are the eigenfunctions of $T_K$ sorted such that the corresponding eigenvalues $\lambda_m$ are non-decreasing, we have

$$\|f_i - f_j\|^2 \geq \beta^2 K, \quad \text{for } i \neq j,$$

and may thus consider a lower bound of the form $A/2 = K\beta^2/8$. Then, since $\|f_i - f_j\|^2 \leq 4\beta^2 K \leq 32\beta^2 \log M$, it suffices to have $16n\beta^2/\sigma_\rho^2 \log M \leq \log M/4$, *i.e.*, $\beta^2 \leq \sigma_\rho^2/64n$ to satisfy the second condition above. Further, in order for all $f_i$ to satisfy the capacity condition, we need $\|T_K^{-r}f_i\|^2 \leq C_*^2$. Note that we have

$$\|T_K^{-r}f_i\|^2 = \beta^2 \sum_{m=1}^K \lambda_m^{-2r} \leq K\beta^2\lambda_K^{-2r},$$

thus, it suffices to take $K\beta^2 \leq C_*^2\lambda_K^{2r}$. Taking a maximal $\beta^2$ under these two conditions, we have the following lower bound on the minimax risk:

$$M_n(\mathcal{F}) \geq \frac{K\beta^2}{8} \geq \frac{1}{8}\min\left\{C_*^2\lambda_K^{2r}, \frac{K\sigma_\rho^2}{64n}\right\}.$$

Using the lower bound $\lambda_K^{2r} \geq (C_0/2)^{2r}K^{-2\alpha r}$, which holds for $K \geq M_0$, and optimizing for $K$ yields $K \approx (C_*^2 C_K^{2r}\sigma_\rho^2 n)^{1/(1+2\alpha r)}$. For $n$ large enough, we have $K \geq M_0$, and the following lower bound holds:

$$M_n(\mathcal{F}) \gtrsim C_*^{\frac{2}{2\alpha r+1}} C_0^{\frac{2r}{2\alpha r+1}}\left(\frac{\sigma_\rho^2}{n}\right)^{\frac{2\alpha r}{2\alpha r+1}}.$$

This matches the upper bound (48) up to absolute constants.

**Tightness of $1/|G|$ for the invariant class.** We now show that the our bound in Theorem 5 which asymptotically shows a $1/|G|n$ instead of $1/n$, is (asymptotically) minimax optimal over the class of *invariant* targets which satisfy assumption (A2).

We consider the same kernel $K_0$ as above, and denote by $K_{G,0}$ its invariant counterpart. It suffices to show that we have the asymptotic expansion

$$\lambda_m(T_{K_{G,0}}) \sim |G|^{-\alpha}C_0 m^{-\alpha} \tag{49}$$

instead of $C_0 m^{-\alpha}$. Indeed, in this case we can construct a kernel $K$ such that its invariant counterpart $K_G$ has eigenvalues upper bounded as $\lambda_m(T_{K_G}) \leq 2C_0|G|^{-\alpha}m^{-\alpha}$ and lower bounded by $(C_0/2)|G|^{-\alpha}m^{-\alpha}$ for $m \geq M_1$ (note that $M_1$ could be chosen large enough so that construction from before for the non-invariant case also applies to the same kernel $K$). Then, applying the same arguments as for the non-invariant case, we obtain the desired minimax-lower bound

$$M_n(\bar{\mathcal{F}}) \gtrsim C_*^{\frac{2}{2\alpha r+1}} C_0^{\frac{2r}{2\alpha r+1}}\left(\frac{\sigma_\rho^2}{|G|n}\right)^{\frac{2\alpha r}{2\alpha r+1}},$$

for $n$ large enough, where $\bar{\mathcal{F}}$ is the class of invariant targets satisfying assumption (A2) with the kernel $K$. This shows that our upper bound is asymptotically tight in a minimax sense for the kernel considered.

We now explain why (49) holds. Let $\mu_k$ denote the Legendre coefficients of $\kappa$ at frequency $k$, and assume $\mu_k \sim C_1 k^{-\beta}$. Recall that we have $N(d,k) \sim C_2 k^{d-2}$, so that $D(k) = \sum_{k' \leq k} N(d,k') \sim C_3 k^{d-1}$ for some $C_3$. Then, when taking eigenvalues with their multiplicity, when $D(k-1) < m \leq D(k)$, the $m$-th eigenvalue $\lambda_m(T_{K_0})$ is $\mu_k$. Asymptotically, we have $k \sim C_3^{\frac{1}{d-1}} m^{\frac{1}{d-1}}$, so that $\lambda_m \sim C_1 C_3^{\frac{-\beta}{d-1}} m^{\frac{-\beta}{d-1}}$, that is, we have $\alpha = \beta/(d-1)$ and $C_0 = C_1 C_3^{-\alpha}$.

Now, since $\frac{\bar{N}(d,k)}{N(d,k)} \to \frac{1}{|G|}$ as $k \to \infty$, we have $\bar{N}(d,k) \sim (C_2/|G|)k^{d-2}$, and $\bar{D}(k) \sim (C_3/|G|)k^{d-1}$. With the same reasoning, this leads to $\lambda_m(T_{K_{G,0}}) \sim C_1(C_3/|G|)^{-\alpha}m^{-\alpha} = C_0|G|^{-\alpha}m^{-\alpha}$, which is the desired constant.

# C  Proofs for Section 4 (Decays of $\gamma_d(k)$)

## C.1  Proof of Proposition 6 (decay of $\gamma_{d,\sigma}(k)$)

The proof of Proposition 6 is technical and relies on identifying and analyzing the singularities in the density $q_\sigma$ of the random variable $Z_\sigma = \langle \sigma \cdot x, x \rangle$, with $x \sim \tau$, using results in [28]. Lemma 15 provides a general integration by parts result which is useful throughout the proof to obtain asymptotic decays from regularity properties. Lemma 17 and Lemma 18 provide asymptotic decays for singularities $\phi(t)$ localized around some $\lambda$ in $(-1, 1)$ and $\{\pm 1\}$, respectively, either through integration by parts or using closed form expressions of certain integrals. Proposition 6 is then proved by appropriately "cancelling" the singularities in $q_\sigma$ using such localized functions $\phi$, as explained in Lemma 19, and applying the integration by parts lemma on the resulting function, which is of higher smoothness and thus leads to faster-decaying terms.

**Lemma 15** (Integration by parts). *Let $g : [-1, 1] \to \mathbb{R}$ be $2s$-times differentiable, with all derivatives bounded on $[-1, 1]$. We then have*

$$\int_{-1}^{1} g(t) P_{d,k}(t)(1-t^2)^{\frac{d-3}{2}} \, dt = \frac{1}{(k(k+d-2))^s} \int_{-1}^{1} \tilde{g}_{d,s}(t) P_{d,k}(t)(1-t^2)^{\frac{d-3}{2}} \, dt, \qquad (50)$$

*where $\tilde{g}_{d,s}$ is a bounded function on $[-1, 1]$.*

*Proof.* We use the following relation, derived in [5, Lemma 4] for a function $f_0$:

$$\int_{-1}^{1} f_0(t) P_{d,k}(t)(1-t^2)^{\frac{d-3}{2}} \, dt = \frac{1}{k(k+d-2)} \Big( -f_0(t)(1-t^2)^{1+\frac{d-3}{2}} P'_{d,k}(t) \Big|_{-1}^{1}$$

$$+ f'_0(t)(1-t^2)^{1+\frac{d-3}{2}} P_{d,k}(t) \Big|_{-1}^{1} + \int_{-1}^{1} f_1(t) P_{d,k}(t)(1-t^2)^{(d-3)/2} \, dt \Big),$$

where $f_1(t) = -f''_0(t)(1-t^2) + (d-1)t f'_0(t)$. Note that the terms in brackets vanish when $f_0$ and $f'_0$ are bounded, and that $f_1$ is $2s - 2$ times differentiable with bounded derivatives if $f_0$ is $2s$ times differentiable. We may thus apply this recursively $s$ times to $f_0 = g$, with

$$f_k(t) = -f''_{k-1}(t)(1-t^2) + (d-1)t f'_{k-1}(t),$$

and we obtain the desired result, with $\tilde{g} = f_s$. $\qquad\square$

**Lemma 16.** *Let $g \in L^\infty([-1, 1])$. It holds that*

$$\left| \int_{-1}^{1} g(t) P_{d,k}(t)(1-t^2)^{(d-3)/2} \, dt \right| \leq 2\pi d^{-1/2} \|g\|_\infty \left( \frac{d}{k} \right)^{(d-2)/2}.$$

*Proof.* By [1, equation (2.117)], we get that

$$|P_{d,k}(t)| \leq \frac{1}{\sqrt{\pi}} \Gamma\left( \frac{d-1}{2} \right) \left( \frac{4}{k(1-t^2)} \right)^{(d-2)/2}$$

$$\leq \frac{1}{\sqrt{\pi}} \left( \frac{d-1}{4} \right)^{(d-3)/2} \left( \frac{4}{k(1-t^2)} \right)^{(d-2)/2}$$

$$\leq \frac{1}{\sqrt{\pi}} \left( \frac{d}{4} \right)^{-1/2} \left( \frac{d}{k(1-t^2)} \right)^{(d-2)/2} \leq 2d^{-1/2} \left( \frac{d}{k(1-t^2)} \right)^{(d-2)/2}.$$

Therefore it follows that

$$\left| \int_{-1}^{1} g(t) P_{d,k}(t)(1-t^2)^{(d-3)/2} \, dt \right| \leq 2d^{-1/2} \|g\|_\infty \left( \frac{d}{k} \right)^{(d-2)/2} \int_{-1}^{1} (1-t^2)^{-1/2} \, dt,$$

which concludes the proof. $\qquad\square$

**Lemma 17** (Decay for $\lambda \in (-1,1)$)**.** *Let $\phi_{\lambda+,\alpha}(t) := (t-\lambda)^\alpha_+ \varphi_{\lambda+,\alpha}(t)$ and $\phi_{\lambda-,\alpha}(t) := (t-\lambda)^\alpha_- \varphi_{\lambda-,\alpha}(t)$, where $\varphi_{\lambda\pm,\alpha} \in C^\infty([-1,1])$ have support $(-1+\epsilon, 1-\epsilon)$ for some $\epsilon > 0$ and take the value $1$ at $t = \lambda$. Then we have that*

$$\left| \int_{-1}^1 \phi_{\lambda\pm,\alpha}(t) P_{d,k}(t) dt \right| \le C(d,\alpha) k^{-d/2-\alpha+3} . \tag{51}$$

*Also let, for $\alpha$ integer, $\phi^*_{\lambda\pm,\alpha}(t) := (t-\lambda)^\alpha_\pm \log|t-\lambda| \varphi^*_{\lambda\pm,\alpha}(t)$, where $\varphi^*_{\lambda\pm,\alpha} \in C^\infty([-1,1])$ have support $(-1+\epsilon, 1-\epsilon)$ for some $\epsilon > 0$ and take the value $1$ at $t = \lambda$. Then we have that*

$$\left| \int_{-1}^1 \phi^*_{\lambda\pm,\alpha}(t) P_{d,k}(t) dt \right| \le C(d,\alpha) k^{-d/2-\alpha+3} . \tag{52}$$

*Proof.* Let $\psi_{\lambda\pm,\alpha}(t) := \phi_{\lambda\pm,\alpha}(t)(1-t^2)^{-(d-3)/2}$. Notice that $\psi_{\lambda+,\alpha}$ satisfies the assumption of Lemma 15 with $2s = 2\lfloor \frac{\alpha}{2} \rfloor \ge \alpha - 2$. Therefore we obtain

$$\left| \int_{-1}^1 \phi_{\lambda+,\alpha}(t) P_{d,k}(t) dt \right| = \left| \int_{-1}^1 \psi_{\lambda+,\alpha}(t) P_{d,k}(t)(1-t^2)^{(d-3)/2} dt \right|$$

$$\le k^{-\alpha+2} \left| \int_{-1}^1 \tilde{\psi}_{\lambda+,\alpha,d}(t) P_{d,k}(t) \left(1-t^2\right)^{(d-3)/2} dt \right|$$

$$\le C(d,\alpha) k^{-\alpha+2-d/2+1} ,$$

where $\tilde{\psi}_{\lambda+,\alpha,d}$ is a bounded function given by Lemma 15, and where we used Lemma 16 to obtain the last inequality. The second inequality follows in the same way, by noticing that the function $\psi^*_{\lambda\pm,\alpha}(t) := \phi^*_{\lambda\pm,\alpha}(t)(1-t^2)^{-(d-3)/2}$ satisfies the assumption of Lemma 15 with $2s = 2\lfloor \frac{\alpha-1}{2} \rfloor \ge \alpha - 2$ (since we assume that $\alpha$ is integer in this case). $\square$

**Lemma 18** (Decay for $\lambda = \pm 1$)**.** *Let $\phi_{1,\alpha,s}(t) := (\frac{1+t}{2})^{\alpha+s-\lfloor\alpha\rfloor}(1-t)^\alpha$, with $\alpha$ non-integer and $s$ integer. Then, $\phi_{1,\alpha,s}$ is $s$ times differentiable at $-1$ and obeys the decay*

$$\left| \int_{-1}^1 \phi_{1,\alpha,s}(t) P_{d,k}(t) dt \right| \le C(d,\alpha,s) k^{-2(\alpha+1)}, \tag{53}$$

*where the constant $C(d,\alpha,s)$ may be different depending on the parity of $k$.*

*Similarly, let $\phi_{-1,\alpha,s}(t) := (\frac{1-t}{2})^{\alpha+s-\lfloor\alpha\rfloor}(t+1)^\alpha$, then $\phi_{-1,\alpha,s}$ is $s$ times differentiable at $1$, and obeys the same decay.*

*Proof.* We begin by evaluating the decay of $\psi_\alpha(t) := (1-t^2)^\alpha$. Following analogous calculations to [5, Lemma 6], we have[4]

$$\int \psi_\alpha(t) P_{d,k}(t) dt \sim C(d,\alpha) k^{-2(\alpha+1)}, \tag{54}$$

for $k$ even, and the integral is equal to zero for $k$ odd. We have

$$C(d,\alpha) = 2^{4\alpha+3} \frac{\omega_{d-2}}{\omega_{d-1}} \frac{\Gamma(\alpha+1)^2}{\Gamma(2\alpha+2)} \frac{\Gamma(\alpha+\frac{3}{2})\Gamma(\frac{d-1}{2})\Gamma(\alpha+\frac{5-d}{2})}{\Gamma(-\frac{1}{2})\Gamma(\alpha+\frac{7-d}{2})\Gamma(-\alpha+\frac{d-5}{2})},$$

where $\omega_{p-1}$ is the surface of $\mathbb{S}^{p-1}$.

Now, let $r := s - \lfloor \alpha \rfloor$, so that we have

$$\phi_{1,\alpha,s}(t) = 2^{-\alpha-r}(1+t)^r \psi_\alpha(t).$$

Let $c_0, \dots, c_r$ denote the coefficients of the degree-$r$ polynomial $p(t) = 2^{-\alpha-r}(1+t)^r$, so that

$$p(t) = 2^{-\alpha-r}(1+t)^r = c_0 + c_1 t + \dots + c_r t^r.$$

---

[4]Note that while Lemma 6 in [5] is stated for $\alpha = \nu + \frac{d-3}{2}$ for $\nu > 0$, the derivation still holds for any $\alpha > -1$.

Using the relation (see, *e.g.*, [14, Proposition 4.21])

$$tP_{d,k}(t) = \frac{k}{2k+d-2}P_{d,k-1}(t) + \frac{k+d-2}{2k+d-2}P_{d,k+1}(t),$$

we may then write

$$p(t)P_{d,k}(t) = \sum_{j=-r}^{r} b_j(k)P_{d,k+j}(t),$$

for some coefficients $b_j(k)$ satisfying $b_j(k) = O(1)$ as $k \to \infty$. Then, we have

$$\int \phi_{1,\alpha,s} P_{d,k}(t)dt = \int \psi_\alpha(t)p(t)P_{d,k}(t)dt$$

$$= \sum_{j=-r}^{r} b_j(k) \int \psi_\alpha(t)P_{d,k+j}(t)dt.$$

When $k \to \infty$, this is a sum of at most $2r + 1 \leq 2s + 1$ terms, each of which decays with $k$ as $k^{-2(\alpha+1)}$ by (54). This yields the result.

The decay for $\phi_{-1,\alpha,s}$ is proved analogously. $\qquad\square$

**Lemma 19** (Cancelling singularities of the density). *Let $q_\sigma$ denote the density of the random variable $Z_\sigma = \langle \sigma \cdot x, x \rangle$, with $\sigma \neq \mathrm{Id}$, and let $\bar{\Lambda}_\sigma$ be the set of eigenvalues of $\bar{A}_\sigma := (A_\sigma + A_\sigma^\top)/2$, where $A_\sigma$ is the permutation matrix of $\sigma$, and denote by $\bar{m}_\lambda$ the multiplicity of $\lambda \in \bar{\Lambda}_\sigma$. Define*

$$\alpha_\lambda = \frac{d - \bar{m}_\lambda}{2} - 1. \tag{55}$$

*There exists constants $\{c_{\lambda,i}\}$ such that the function defined by*

$$\tilde{q}_\sigma = \sum_{\lambda \in \Lambda_\sigma \setminus \{1, \lambda_{\min}\}} \sum_{i=0}^{\lceil d+1-\alpha_\lambda \rceil} (c_{\lambda+,i}\phi_{\lambda+,\alpha_\lambda+i} + c_{\lambda-,i}\phi_{\lambda-,\alpha_\lambda+i} + c_{\lambda+,i}^*\phi_{\lambda+,\alpha_\lambda+i}^* + c_{\lambda-,i}^*\phi_{\lambda-,\alpha_\lambda+i}^*)$$

$$+ \mathbf{1}\{\lambda_{\min} > -1\} \sum_{i=0}^{\lceil d+1-\alpha_{\lambda_{\min}} \rceil} (c_{\lambda_{\min}+,i}\phi_{\lambda_{\min}+,\alpha_{\lambda_{\min}}+i} + c_{\lambda_{\min}+,i}^*\phi_{\lambda_{\min}+,\alpha_{\lambda_{\min}}+i}^*)$$

$$+ \sum_{\lambda \in \Lambda_\sigma \cap \{\pm 1\}} \sum_{i=0}^{\lceil d+1+\frac{d-3}{2}-\alpha_\lambda \rceil} c_{\lambda,i}\phi_{\lambda,\alpha_\lambda+i,\lceil d+1+\frac{d-3}{2}\rceil}$$

*satisfies that $t \mapsto (q_\sigma(t) - \tilde{q}_\sigma(t))(1-t^2)^{-\frac{d-3}{2}}$ admits $d+1$ bounded derivatives on $[-1,1]$.*

*Proof.* Note that we have

$$\langle \sigma \cdot x, x \rangle = \frac{1}{2}(\langle A_\sigma x, x \rangle + \langle x, A_\sigma x \rangle) = \langle \bar{A}_\sigma x, x \rangle,$$

where $\bar{A}_\sigma = \frac{1}{2}(A_\sigma + A_\sigma^\top)$ is symmetric and thus has real eigenvalues. When $A_\sigma$ is a permutation matrix, these eigenvalues are in $[-1,1]$, as the real part of complex roots of unity.

We then identify the singularities of the density $q_\sigma$, which are the same as those of the cumulative distribution function, up to one fewer degree of smoothness. Such singularities are shown in the following lemma, proved in [28].

**Lemma 20** ([28]). *Consider $\sigma \in G$ as above, and let $\bar{\Lambda}_\sigma$ be the set of eigenvalues of $\bar{A}_\sigma$. For each $\lambda \in \bar{\Lambda}_\sigma$, we denote by $\bar{m}_\lambda$ its multiplicity. Then the cumulative distribution function $Q_\sigma$ of $Z_\sigma = \langle \sigma \cdot x, x \rangle$ takes the form*

$$Q_\sigma(t) = \varphi(t) + \sum_{\lambda \in \bar{\Lambda}_\sigma} g_\lambda(t)$$

*where $\varphi$ is analytic and*

- $g_\lambda(t) = |t - \lambda|(t - \lambda)^{\frac{d-\bar{m}_\lambda}{2}-1}\varphi_\lambda^1(t) + (t - \lambda)^{\frac{d-\bar{m}_\lambda}{2}}\log(|t-\lambda|)\,\varphi_\lambda^2(t)$ *if* $d - \bar{m}_\lambda$ *is even,*

- $g_\lambda(t) = (t - \lambda)_+^{\frac{d-\bar{m}_\lambda}{2}}\varphi_\lambda^1(t) + (\lambda - t)_+^{\frac{d-\bar{m}_\lambda}{2}}\,\varphi_\lambda^2(t)$ *if* $d - \bar{m}_\lambda$ *is odd,*

*for some* $\varphi_\lambda^1, \varphi_\lambda^2$ *analytic. Further, the term involving* $\log(|t - \lambda|)$ *only appears for* $\lambda \in (-1, 1)$.

In particular, it follows from this lemma by differentiation that we may write

$$q_\sigma(t) = \tilde{\varphi}(t) + \sum_{\lambda \in \bar{\Lambda}_\sigma} \tilde{g}_\lambda(t),$$

with $\tilde{\varphi}$ analytic and

- $\tilde{g}_\lambda(t) = (t - \lambda)_+^{\frac{d-\bar{m}_\lambda}{2}-1}\tilde{\varphi}_{\lambda,1}(t) + (t - \lambda)_-^{\frac{d-\bar{m}_\lambda}{2}-1}\tilde{\varphi}_{\lambda,2}(t) + (t - \lambda)_+^{\frac{d-\bar{m}_\lambda}{2}-1}\log(|t-\lambda|)\tilde{\varphi}_{\lambda,3}(t) +$
  $(t - \lambda)_-^{\frac{d-\bar{m}_\lambda}{2}-1}\log(|t-\lambda|)\tilde{\varphi}_{\lambda,4}(t)$, if $d - \bar{m}_\lambda$ is even

- $\tilde{g}_\lambda(t) = (t - \lambda)_+^{\frac{d-\bar{m}_\lambda}{2}-1}\tilde{\varphi}_{\lambda,1}(t) + (t - \lambda)_-^{\frac{d-\bar{m}_\lambda}{2}-1}\tilde{\varphi}_{\lambda,2}(t)$, if $d - \bar{m}_\lambda$ is odd,

where $\tilde{\varphi}_{\lambda,i}$ are analytic for $i \in [4]$.

The result then follows by appropriately "cancelling" those singularities up to order $d$ using the simple functions $\phi$ introduced in the previous lemmas, and noting that for singularities at $\pm 1$, we require an additional $(d - 3)/2$ degrees of smoothness, so that we may divide by the weight function $(1 - t^2)^{(d-3)/2}$.

For instance, for $\lambda \in (-1, 1)$, an appropriate exponent $\alpha$ and an analytic $\varphi$, we may write

$$(t - \lambda)_+^\alpha \varphi(t) = (t - \lambda)_+^{\frac{d-\bar{m}_\lambda}{2}-1}(c_0 + (t - \lambda)\psi(t)), \tag{56}$$

with $\psi$ analytic. Then for $\phi_{\lambda+,\alpha}$ as in Lemma 17, we have

$$(t - \lambda)_+^\alpha \varphi(t) - c_0\phi_{\lambda+,\alpha}(t) = (t - \lambda)_+^{\alpha+1}\tilde{\varphi}(t),$$

with $\tilde{\varphi}$ analytic. We may then repeat this process with functions $\phi_{\lambda+,\alpha+1}$, $\phi_{\lambda+,\alpha+2}$, etc., to finally obtain that

$$(t - \lambda)_+^\alpha \varphi(t) - \sum_{i=0}^{\lceil d+1-\alpha \rceil} c_i\phi_{\lambda+,\alpha+i}(t)$$

is $d$ times differentiable, as desired. A similar reasoning can be applied for other types of singularities. For the terms involving $\log(|t - \lambda|)$, which only appear for $\lambda \in (-1, 1)$, note that the corresponding exponent $\alpha_\lambda$ is integer since $d - \bar{m}_\lambda$ is even, so that Lemma 17 applies.

$\square$

We are now ready to state the proof of Proposition 6.

*Proof of Proposition 6.* Let $q_\sigma$ be the density of $\langle \sigma \cdot x, x \rangle$, and let $\tilde{q}_\sigma$ be as in Lemma 19. By Lemma 19 and Lemma 15, we have

$$\int_{-1}^1 (q_\sigma(t) - \tilde{q}_\sigma(t))P_{d,k}(t)dt = \int_{-1}^1 \frac{q_\sigma(t) - \tilde{q}_\sigma(t)}{(1 - t^2)^{\frac{d-3}{2}}}P_{d,k}(t)(1 - t^2)^{\frac{d-3}{2}}dt \leq Ck^{-d},$$

where we have bounded the integral on the r.h.s. of (50) by a constant. Renaming the terms in $\tilde{q}$ as $\tilde{q}_\sigma = \sum_i c_i q_i$, where each $q_i$ is as in Lemma 17 or Lemma 18, we have

$$\gamma_{d,\sigma}(k) \leq \sum_i c_i \int q_i P_{d,k} + O(k^{-d}).$$

Now, note that the eigenvalues of $\bar{A}_\sigma = \frac{1}{2}(A_\sigma + A_\sigma^\top)$ are the real parts of the complex eigenvalues of the permutation matrix $A_\sigma$. Since $A_\sigma$ is real, its complex eigenvalues come in conjugate pairs, so that

for any $\lambda \in \Lambda_\sigma$ with $\lambda \notin \mathbb{R}$, we have $m_\lambda = \bar{m}_{Re(\lambda)}/2$, where $\bar{m}$ are the multiplicities of Lemma 19. Note that in the case of permutations, the eigenvalues of $A_\sigma$ are roots of unity, so that we have

$$m_\lambda = \begin{cases} \bar{m}_\lambda, & \text{if } \lambda \in \{\pm 1\} \\ \bar{m}_{Re(\lambda)}/2, & \text{otherwise.} \end{cases}$$

The result then follows by applying the decays given by Lemma 17 and Lemma 18 to each component $q_i$ with the appropriate $\alpha_\lambda$, and focusing on the leading-order terms. Namely, for $\lambda \in \{\pm 1\}$, by Lemma 18, the leading order term in $\gamma_{d,\sigma,\lambda}(k)$ has decay

$$k^{-2(\alpha_\lambda + 1)} = k^{-2(\frac{d - \bar{m}_\lambda}{2} - 1 + 1)} = k^{-d + m_\lambda},$$

while for $\lambda \notin \{\pm 1\}$, we have $Re(\lambda) \in (-1, 1)$, hence using (55) and $\bar{m}_{Re(\lambda)} = 2m_\lambda$, we have $\alpha_{Re(\lambda)} = \frac{d}{2} - m_\lambda - 1$, so that the decay, by Lemma 17, is upper bounded by

$$k^{-\frac{d}{2} - \alpha_{Re(\lambda)} + 3} \leq k^{-d + m_\lambda + 4}.$$

This concludes the proof. $\qquad\square$

## C.2  Proof of Corollary 7 (leading order of $\gamma_{d,\sigma}$)

*Proof.* By Proposition 6, we get that

$$\gamma_{d,k}(\sigma) \lesssim k^{-d+s}$$

where $s = \max_{\lambda \in \Lambda_\sigma} \{m_\lambda + 4 \cdot \mathbf{1}(|\lambda| < 1)\}$. Notice that, for any permutation $\sigma$, it holds $m_1 \geq m_\lambda$. Therefore, we have

$$s \leq m_1 + 4 \cdot \mathbf{1}\{\exists |\lambda| < 1 \ : \ m_1 < m_\lambda + 4\}.$$

Now, if $m_1 < m_\lambda + 4$ for some $|lambda| < 1$, since $m_1 + m_\lambda \leq d$, it must hold $2m_1 \leq d + 3$, or, equivalently, $m_1 \leq d/2 + 3/2$. It follows that

$$s \leq \begin{cases} m_1 & \text{if } m_1 > (d+3)/2 , \\ d/2 + 5.5 & \text{otherwise.} \end{cases}$$

This concludes the proof, since $m_1 = c(\sigma)$. $\qquad\square$

## C.3  Proof of Corollary 8 (different upper bounds using permutation statistics)

*Proof.* For all $\sigma \in G \setminus \zeta(G, s)$, it holds $c(\sigma) \leq s$ and

$$\gamma_{d,\sigma}(k) \lesssim k^{-d + \eta_\sigma} ,$$

with

$$\eta_\sigma = c(\sigma) \cdot \mathbf{1}\left(c(\sigma) > (d+3)/2\right) + \left(\frac{d}{2} + 6\right) \mathbf{1}\left(c(\sigma) \leq (d+3)/2\right) .$$

In particular, we have

$$\gamma_d(k) = \frac{\zeta(G, s)}{|G|} + O\left(k^{-d+\eta}\right)$$

where

$$\eta = \max_{c(\sigma) \leq s} \eta_\sigma .$$

Denote $s^*(s) = \max_{\sigma \in G \setminus \zeta(G,s)} c(\sigma)$. If $s^*(s) \geq d/2 + 6$, then it holds that $s^*(s) = \eta$. Otherwise, $c(\sigma) \leq d/2 + 7$ for any $\sigma$ such that $c(\sigma) \leq s$, which implies that $s \leq d/2 + 6$. It follows that

$$\eta = \max\{s^*(s), \ d/2 + 6\} \leq \max\{s, \ d/2 + 6\}.$$

$\qquad\square$

## C.4 Details on Example 11 (full permutation group)

The number of permutations in $G = S_d$ which fix exactly $n$ elements is given by $\binom{n}{k}\,!(n-k)$, where $!m$ denotes the $m$-th subfactorial:

$$!m := \left\lfloor \frac{m!+1}{e} \right\rfloor \leq \frac{2m!}{e}\,.$$

It follows that

$$\frac{\xi(G,s)}{|G|} = \frac{1}{d!}\sum_{k=s+1}^{d} \frac{d!}{(d-k)!k!}\,!(d-k) \leq \frac{2}{e}\sum_{k=s+1}^{d}\frac{1}{k!}$$

$$\leq \frac{2}{e(s+1)!}\sum_{k=s+1}^{d}\frac{1}{(s+2)^{k-(s+1)}} = \frac{2}{e(s+1)!}\frac{1-\frac{1}{(s+2)^{d-s}}}{1-\frac{1}{s+2}}$$

$$\leq \frac{2(s+2)}{e(s+1)}\frac{1}{(s+1)!} \leq \frac{2}{(s+1)!}\,.$$

# D   Proofs for Section 5

## D.1   Proof of Lemma 12 (spectral properties of smoothing operator $S_G$)

*Proof.* As in the invariant case, we note that for any degree $k$, the space $V_{d,k}$ of spherical harmonics of degree $k$ is stable by $S_G$, i.e., $S_G V_{d,k} \subset V_{d,k}$. Since $S_G$ is self-adjoint, we may then find an orthonormal basis of such spherical harmonics, which we denote $\overline{Y}_{k,j}$, for $j = 1,\ldots,N(d,k)$, such that the restriction of $S_G$ to $V_{d,k}$ is diagonal, and we have $S_G\overline{Y}_{k,j} = \lambda_{k,j}\overline{Y}_{k,j}$, with $\lambda_{k,j} \geq 0$.

It remains to show (20). Define the operator $\Pi_k f = \mathbb{E}_y\left[P_{d,k}(\langle\cdot,y\rangle)f(y)\right]$. $S_G\Pi_k$ is then an integral operator with kernel

$$H(x,y) = \sum_{\sigma\in G} h(\sigma)P_{d,k}(\langle\sigma\cdot x,y\rangle). \tag{57}$$

Since $\overline{Y}_{k,j}$, $j = 1,\ldots,N(d,k)$ forms an orthonormal basis of $V_{d,k}$, by the addition formula of spherical harmonics, we have

$$\Pi_k = \frac{1}{N(d,k)}\sum_{j=1}^{N(d,k)} \overline{Y}_{k,j}\overline{Y}_{k,j}^*.$$

It follows that

$$S_G\Pi_k = \frac{1}{N(d,k)}\sum_{j=1}^{N(d,k)} S_G\overline{Y}_{k,j}\overline{Y}_{k,j}^* = \frac{1}{N(d,k)}\sum_{j=1}^{N(d,k)} \lambda_{k,j}\overline{Y}_{k,j}\overline{Y}_{k,j}^*.$$

This implies that the kernel $H$ of the operator $S_G\Pi_k$ can also be expressed as

$$H(x,y) = \frac{1}{N(d,k)}\sum_{j=1}^{N(d,k)} \lambda_{k,j}\overline{Y}_{k,j}(x)\overline{Y}_{k,j}(y). \tag{58}$$

Fixing $y = x$ and taking expectations over $x \sim \tau$ in both (57) and (58) proves the equality.

$\square$

## D.2   Proof of Theorem 13 (generalization with geometric stability)

The proof of the theorem is analogous to that of Theorem 5, replacing the control on $\mathcal{N}_{K_G}(\lambda)$ with that of Lemma 21 below, which provides an extension of Lemma 4 to generic smoothing operators, at the cost of a weaker constant $\nu_d(\ell)^{1/\alpha}$ instead of $\nu_d(\ell)$. The remark on the constant $C_4$ being potentially smaller for the kernel $K$ stems from the fact that we no longer have equal approximation errors for the two kernels, and that the quantity $C_{f^*}$ in (40) in this case is the one given by the source

condition (A2) instead of (A6), which is smaller, as we now show. Indeed, note that if $f^* = S_G^r T_K^r g$, then we have $f^* = T_K^r(S_G^r g) = T_K^r \tilde{g}$ with $\tilde{g} = S_G^r g$, since $S_G$ and $T_K$ are diagonalized in the same basis and hence commute. The result follows by noting that we have $\|\tilde{g}\|_{L^2(d\tau)} \leq \|g\|_{L^2(d\tau)}$, since $\|S_G^r\| \leq \|S_G\|^r \leq 1$ (indeed we have the operator norm bound $\|S_G\| \leq 1$, which follows from a simple triangle inequality).

**Lemma 21** (Degrees of freedom for $K_G$ with stability.)**.** *Assume (A5). We have*

$$\mathcal{N}_K(\lambda) \leq C_K \lambda^{-1/\alpha},$$

*and for any $\ell \geq 0$, we have*

$$\mathcal{N}_{K_G}(\lambda) \leq D(\ell) + \nu_d(\ell)^{1/\alpha} C_K \lambda^{-1/\alpha}, \tag{59}$$

*with the same constant $C_K$.*

*Proof.* The first statement is a standard consequence of Assumption (A5). Namely, if $\xi_m$ denote the eigenvalues of $T_K$ (namely, the same as $\mu_k$ counted with their multiplicities) and $\xi_m \leq C(m+1)^{-\alpha}$, we have

$$\begin{aligned}
\mathcal{N}_K(\lambda) &= \sum_{m \geq 0} \frac{\xi_m}{\xi_m + \lambda} \\
&\leq \sum_{m \geq 0} \frac{1}{1 + \lambda C^{-1}(m+1)^\alpha} \\
&\leq \int_0^\infty \frac{dt}{1 + \lambda C^{-1} t^\alpha} \\
&\leq \frac{C^{1/\alpha} \lambda^{-1/\alpha}}{\alpha} \int_0^\infty \frac{u^{1/\alpha - 1} du}{1 + u} = C_K \lambda^{-1/\alpha},
\end{aligned}$$

with $C_K := \frac{C^{1/\alpha}}{\alpha} \int_0^\infty \frac{u^{1/\alpha - 1} du}{1+u}$.

We now write

$$\begin{aligned}
\mathcal{N}_{K_G}(\lambda) &= \sum_{k \geq 0} \sum_{j=0}^{N(d,k)} \frac{\lambda_{k,j} \mu_k}{\lambda_{k,j} \mu_k + \lambda} \\
&= \sum_{k \geq 0} \sum_j \frac{\lambda_{k,j}}{\lambda_{k,j} + \lambda \mu_k^{-1}} \\
&\leq \sum_k N(d,k) \frac{\bar{\lambda}_k}{\bar{\lambda}_k + \lambda \mu_k^{-1}} \quad \text{(by Jensen's inequality, with } \bar{\lambda}_k = N(d,k)^{-1} \sum_j \lambda_{k,j}) \\
&\leq \sum_k N(d,k) \frac{\bar{\lambda}_k \mu_k}{\bar{\lambda}_k \mu_k + \lambda},
\end{aligned}$$

We may then write, for some $\ell \geq 1$,

$$\mathcal{N}_{K_G}(\lambda) \leq D(\ell) + \sum_{k \geq 0} \frac{\bar{\mu}_k}{\bar{\mu}_k + \lambda},$$

where

$$\bar{\mu}_k = \begin{cases} \bar{\lambda}_k \mu_k, & \text{if } k \geq \ell \\ 0, & \text{o/w.} \end{cases}$$

Note that for $k \geq \ell$, we have $\bar{\lambda}_k = \frac{\sum_j \lambda_{k,j}}{N(d,k)} = \gamma_d(k) \leq \nu_d(\ell)$, and the same holds trivially for $k < \ell$. Then, writing $\bar{\xi}_m$ the collections of $\bar{\mu}_k$ counted with multiplicities, we may write $\bar{\xi}_m \leq \nu_d(\ell) C(m+1)^{-\alpha}$, with the same constant $C$ as in (A5). Repeating the argument above for bounding $\mathcal{N}_{K_G}(\lambda)$ in terms of $\lambda^{-1/\alpha}$ then yields the result. $\qquad\square$

### D.3 Proof of Proposition 14 (upper bound on $\gamma_d(k)$ for deformations)

*Proof.* We first show that $\Phi_2$ is stable under inversion, and later proceed to study lower bounds on its number of elements, and cycle statistics.

**Step 1: $\Phi_2^{-1} = \Phi_2$.** Let us first establish that $\Phi_2$ is closed under inversion.

First observe that

$$\Phi_2 = \{\sigma; |\sigma(u+1) - \sigma(u) - 1| \leq 2 \ \forall \ u\} \,. \tag{60}$$

The inclusion LHS $\subseteq$ RHS is immediate by definition. The reverse inclusion is obtained by the triangle inequality, by observing that if $u < \tilde{u} < u'$, then

$$
\begin{aligned}
|\sigma(u) - \sigma(u') - (u - u')| &= |\sigma(u) - \sigma(\tilde{u}) - (u - \tilde{u}) + \sigma(\tilde{u}) - \sigma(u') - (\tilde{u} - u')| \\
&\leq |\sigma(u) - \sigma(\tilde{u}) - (u - \tilde{u})| + |\sigma(\tilde{u}) - \sigma(u') - (\tilde{u} - u')| \,,
\end{aligned}
$$

so by induction if the condition holds for small pairs $(u, \tilde{u})$, $(\tilde{u}, u')$ it extends to all pairs $(u, u')$.

We directly verify from (60) that $\sigma \in \Phi_2$ iff it holds

$$\forall \ u \,, \ \sigma(u+1) = \sigma(u) + \{3, 2, 1, -1\} \,, \tag{61}$$

since we need to have $\sigma(u) \neq \sigma(u')$ whenever $u \neq u'$.

Let now $\tilde{u} = \sigma(u)$, so $\sigma^{-1}(\tilde{u}) = u$. We will show that $\sigma^{-1}$ also verifies (61). We want to enumerate all possible $u'$ so that $\sigma(u') = \tilde{u} + 1$. Clearly $\sigma^{-1}(\tilde{u} + 1) \neq \sigma^{-1}(\tilde{u})$ so $u' \neq u$.

Suppose by contradiction that $u' < u - 1$. Note that we must have $\sigma(u' + 1) \geq \tilde{u} + 2$ since $\tilde{u}$ and $\tilde{u} + 1$ already have pre-images (namely $u$ and $u'$), and smaller values would violate $\sigma(u' + 1) - \sigma(u') \in \{3, 2, 1, -1\}$. Similarly $\sigma(u' + s) \geq \tilde{u} + 2$ for all $s = 2, \ldots, u - u' - 1$, since otherwise we would need a step $\sigma(u' + s + 1) - \sigma(u' + s) \leq -3$, which is ruled out by (61). Then it must be that $\sigma(u) - \sigma(u-1) = \tilde{u} - \sigma(u-1) \leq -2$, which is a contradiction. We have thus shown $u' \geq u - 1$.

Similarly, let us show $u' \leq u + 3$. Assume, by contradiction that $u' > u + 3$. Note first that the only way to have $\sigma(u + s) < \tilde{u}$ for some $s \in [0, u' - u]$ is to only have $\sigma(u + 1) = \tilde{u} - 1$, and $\sigma(u + s) \geq \tilde{u} + 2$ otherwise. Indeed, values smaller than $\tilde{u}$ must happen just following $u$ in order to allow decreasing by 1, and having additional negative steps after $u + 1$ (*e.g.*, $\sigma(u + 2) = \tilde{u} - 2$) would require a step $\sigma(u + s + 1) - \sigma(u + s) > 3$ for some $s \in [2, u' - u]$ (since the values $\tilde{u} - 1, \tilde{u}, \tilde{u} + 1$ already have pre-images, given by $u + 1, u, u'$, respectively), which is a contradiction. Then, if $\sigma(u+1) = \tilde{u} - 1$, we must have $\sigma(u+2) = \tilde{u} + 2$ (since we cannot have longer steps), which implies $\sigma(u' - 1) \geq \tilde{u} + 3$ and thus $\sigma(u') - \sigma(u' - 1) \leq -2$, which is a contradiction. Alternatively, we must have $\sigma(u + s) \geq \tilde{u} + 2$ for all $s \in [1, u' - u - 1]$. This implies $\sigma(u' - 1) = \tilde{u} + 2$, $\sigma(u' - 2) = \tilde{u} + 3$, and more generally $\sigma(u' - t) = \tilde{u} + t + 1$, since these are the only allowed steps to obtain $\sigma(u') = \tilde{u} + 1$. Then, we have $\sigma(u + 1) = \sigma(u' - (u' - u - 1)) = \tilde{u} + (u' - u) > \tilde{u} + 3$, which is in contradiction with $\sigma(u+1) - \sigma(u) \leq 3$. We have thus proved $u' \leq u + 3$. We thus have that $\sigma^{-1}$ satisfies (61), which shows $\Phi_2^{-1} = \Phi_2$.

**Step 2: Lower bound on $|\Phi_2|$.** Denote as before $\sigma(u) = \tilde{u}$. By denoting $\Delta_v = \sigma(v) - v$ for arbitrary $v$, observe that $\Delta_{u'} - \Delta_{u'-1} \in \{2, 1, 0, -1\}$. Similarly we define $\Gamma_{\tilde{u}} := \sigma^{-1}(\tilde{u}) - \tilde{u}$. By the previous argument, we also have $\Gamma_{\tilde{u}+1} = \Gamma_{\tilde{u}} + \{2, 1, 0, -1\}$. Fix an arbitrary $u_0$, say $u_0 = 1$ and consider the subset of $\Phi_2$ given by

$$\Phi_2^b = \{\sigma \in \Phi_2; \sigma(u_0) = u_0, \sigma(u_0 - 1) = u_0 - 1\} \,.$$

$\Phi_2^b$ thus contains permutations with 'fixed' boundary conditions. For $\sigma \in \Phi_2^b$, the boundary condition prevents $\Delta_{u_0+1} < \Delta_{u_0}$, so we identify the following possible cases:

- **1-block:** $\Delta_{u_0+1} = \Delta_{u_0}$. In this case, $\sigma(u_0 + 1) = \sigma(u_0) + 1$.

- **2-block:** $\Delta_{u_0+1} = \Delta_{u_0} + 1$. This implies $\Gamma_{\tilde{u}_0+2} = \Gamma_{\tilde{u}_0} - 1$, which in turn implies $\Gamma_{\tilde{u}_0+1} = \Gamma_{\tilde{u}_0} + 1$, and finally $\Delta_{u_0+2} = \Delta_{u_0+1} - 2$. In summary, $\sigma(u_0 + 1) = \sigma(u_0) + 2$ and $\sigma(u_0 + 2) = \sigma(u_0) + 1$.

- **3-block:** $\Delta_{u_0+1} = \Delta_{u_0} + 2$. This implies $\Gamma_{\tilde{u}_0+3} = \Gamma_{\tilde{u}_0} - 2$, which necessarily implies $\Gamma_{\tilde{u}_0+1} = \Gamma_{\tilde{u}_0} + 2$, $\Gamma_{\tilde{u}_0+2} = \Gamma_{\tilde{u}_0+1} - 2$ and $\Gamma_{\tilde{u}_0+3} = \Gamma_{\tilde{u}_0+2} - 2$. This corresponds to $\sigma(u_0 + 1) = \sigma(u_0) + 3$, $\sigma(u_0 + 2) = \sigma(u_0) + 2$ and $\sigma(u_0 + 3) = \sigma(u_0) + 1$.

So an element of $\Phi_2^b$ can be constructed sequentially by assembling three possible 'blocks' $B_i$ of size $i = \{1, 2, 3\}$. Moreover, we verify immediately that the following transitions are admissible:

$$B_1 \to B_{\{1,2,3\}} \ , \ \ B_2 \to B_{\{1,2\}} \ , \ \ B_3 \to B_1 \ .$$

Thus, by denoting $\mathcal{B}(m; B_i)$ the number of permutations in $\Phi_2^b$ restricted to their first $m$ elements, and which that start (after $u_0$) with a block of type $B_i$, we have the following recursion:

$$
\begin{array}{rcl}
\mathcal{B}(m; B_1) & = & \mathcal{B}(m - 1; B_1) + \mathcal{B}(m - 1; B_2) + \mathcal{B}(m - 1; B_3) \\
\mathcal{B}(m; B_2) & = & \mathcal{B}(m - 2; B_2) + \mathcal{B}(m - 2; B_1) \\
\mathcal{B}(m; B_3) & = & \mathcal{B}(m - 3; B_1) \ ,
\end{array}
\tag{62}
$$

with $\mathcal{B}(i; B_i) = 1$. Let $F_i(z) := \sum_{m \geq 0} \mathcal{B}(m, B_i) z^m$ be the generating function associated to each of the above sequences. We have

$$F_1(z) = z^{-1}(F_1(z) + F_2(z) + F_3(z)) \ , \ \ F_2(z) = z^{-2}(F_1(z) + F_2(z)) \ , \ \ F_3(z) = z^{-3} F_1(z) \ .$$

By substituitng $F_2, F_3$ into the first equation, we obtain

$$F_1(z)(1 - z^{-1} - z^{-3}(1 - z^{-2})^{-1} - z^{-4}) = 0 \ ,$$

so $F_1$ has a pole at $\tau \approx 1.714$, the solution of the associated characterstic equation $z = 1 + \frac{1}{z^2 - 1} + \frac{1}{z^3}$. Moreover, this pole is also present in $F_2$ and $F_3$. This shows that $\mathcal{B}(m, B_i) \asymp C_i \tau^m$, and hence

$$|\Phi_2| \geq |\Phi_2^b| = \sum_{i=1}^3 \mathcal{B}(d; B_i) = \Theta(\tau^d) \ .$$

**Step 3: cycle statistics.** Let us now compute a bound for $\gamma_d(k)$ using Corollary 8. Let

$$\xi(\Phi_2, n; d) = \{\sigma \in \Phi_2 \ : \ \mathrm{Fix}(\sigma) \geq n\}$$

denote the set of elements of $\Phi_2$ that fix at least $n$ positions, set $n = (1 - \eta)d$, and assume $\eta < 1/2$. Observe that this necessarily implies that two consecutive indices, say $u_0$ and $u_0 - 1$, are fixed, by the pigeonhole principle. Thus

$$\xi(\Phi_2, n) \subset \Phi_2^b \ ,$$

and we can use the characterisation of elements in $\Phi_2^b$. We have

$$|\xi(\Phi_2, (1 - \eta)d; d)| \leq \sum_{n'=(1-\eta)d}^d \binom{d}{d - n'} |\mathcal{B}(n'; B_1) + \mathcal{B}(n'; B_2) + \mathcal{B}(n'; B_3)|$$

$$\leq C\tau^{\eta d} \sum_{n'=(1-\eta)d}^d \binom{d}{d - n'} \leq C\left(e\eta^{-1}\right)^{\eta d} \tau^{\eta d} \ ,$$

where $C$ is an abolute constant. Finally, from Example 11, we have

$$n < c(\sigma) \Rightarrow 2n - d < \mathrm{Fix}(\sigma) \ ,$$

thus $\zeta(\Phi_2, n) \leq \xi(\Phi_2, 2n - d)$. By picking $n = (1 - \eta)d$ with $\eta < 1/4$, we have $2n - d = (1 - 2\eta)d > d/2$ and

$$
\begin{array}{rcl}
\gamma_d(k) & \leq & \dfrac{\xi(\Phi_2, (1 - 2\eta)d)}{|\Phi_2|} + O\left(k^{-d + \max(n, d/2 + 7)}\right) \\[2ex]
& \leq & C(e(2\eta)^{-1})^{2\eta d} \tau^{(2\eta - 1)d} + O\left(k^{-d + \max(n, d/2 + 7)}\right) \ . \\[2ex]
& = & C\left(\dfrac{e^{2\eta}}{(2\eta)^{2\eta} \tau^{1 - 2\eta}}\right)^d + O\left(k^{-\eta d}\right) \ .
\end{array}
$$

$$\tag{63}$$
$$\tag{64}$$
$$\tag{65}$$

When $2\eta < 0.15$, we verify that $\frac{e^{2\eta}}{(2\eta)^{2\eta} \tau^{1 - 2\eta}} < 1$.

$\square$