# OpenReview forum: "On the Sample Complexity of Learning under Geometric Stability"
_NeurIPS.cc/2021/Conference — NeurIPS 2021 Poster_

### Official Review · Reviewer_ey2p · 2021-07-11

**Rating:** 7
**Confidence:** 3

**Summary:**

This paper considers the problem of learning a function $ f_\ast \colon \mathbb{S}^{d-1} \rightarrow R $ from i.i.d. data points $(x_1, y_1), \dots (x_n, y_n)$ drawn from an unknown distribution $\rho$ such that the marginal distribution of $x_i$ is over the sphere $S^{d-1}$ and $\mathbb{E}(y \mid x) = f_*(x)$. Further, they assume that $f_*$ is invariant under the action of a subgroup $G$ of $S_d$ (the group of permutations in $d$ elements), i.e.,
$$ f(g \cdot x) = f(x) \qquad \text{for all }g \in G, x \in \mathbb{S}^{d-1}.$$
The main research question of the paper is whether one can derive better generalization error bound for methods that incorporate the invariance prior. They answer this question in the affirmative for Kernel ridge regression (KRR). For a given positive definite kernel $K$ and its associated Reproducing Kernel Hilbert Space (RKHS) $H_K,$ they study the standard KRR estimator
$$
\widehat f_{\lambda,K} = \text{argmin} \frac{1}{n}\sum_{i=1}^n (f(x_i)-y_i)^2 + \lambda \|f\|^2_{H_K}
$$
and consider two kernels:
$$
K(x, x') = \kappa(\langle x, x'\rangle), \quad K_G(x, x') = \frac{1}{|G|} \sum_{g \in G} K(g \cdot x, x') \quad \text{for }x, x' \in \mathbb{S}^{d-1}
$$
with $\kappa(u) \leq 1$. For both of these kernels, the authors established sample complexity bounds for $\mathbb{E} (\widehat f_{\lambda, K} - f_*)$ under reasonable assumptions. The bounds suggest that when $n \rightarrow \infty$, the risk given by $K_G$ is better by a factor of $\frac{1}{|G|}$.

The authors extend the results to the case where $f^*$ is stable under small geometrical changes on the input, albeit with weaker statements.


**Limitations And Societal Impact:**

Limitations are stated in the last section. This paper studies a theoretical question with no explicit societal impact.

**Main Review:**

This paper studies a relatively new and exciting question. It amasses theoretical evidence to support the use of learning methods that leverage invariance/symmetry. Unlike previous works, this paper studies a broader class of (nonsmooth) functions and assumes that the dimension is fixed and the number of samples grows.

The more substantial results derived in the paper suggest that incorporating invariance into the kernel gives gains of at most a constant factor (determined by the size of the group), and it is indicated in Section 3 that this is the best one can hope for. However, I believe the manuscript would benefit from a more in-depth explanation of optimality in this setting.

Although the topic is highly technical, the paper is clear and reasonably easy to follow. The proofs seem to be mathematically rigorous. However, I did not check all the details. Overall the article is an interesting and well-written piece of work. Below I include a list of suggestions.

- The theorem statements give upper bounds for both kernels. The results would be stronger if the statement for $\widehat f_{\lambda, K}$ included a lower bound.
- The estimator is not guaranteed to be invariant under the action of the group. Do the authors think that this could bring additional gains? Elaborate on the main difficulties in developing theory where the invariance is enforced in the estimator f rather than the kernel.
- L68 sudies -> studies.
- L95 ",which" -> ", which"
- L130 ". leading" -> ". Leading".
- L141 rigression -> regression.
- L233 traslations -> translations.


**Time Spent Reviewing:**

8

---

> ### Author Response · Authors · 2021-08-10
> **Response to Reviewer ey2p**
>
> Thank you for the thorough review and helpful comments.
>
> *Lower bound.* Thank you for raising this point. See the paragraph "optimality" in our response to reviewer BHMR for more discussion.
>
> *Invariant estimator*. Note that when using the invariant kernel, the estimator *is* invariant under the action of the group (did we understand your question correctly?). When using the non-invariant kernel, one may indeed consider other ways to make the estimator invariant, e.g. by taking the symmetrization of the standard KRR estimator (with the non-invariant kernel), however it is shown in [Mei-Misiakiewicz-Montanari '21, Section 5.1] that such an estimator performs worse than using the invariant kernel.
>
> *Typos.* Thank you for pointing these out, we will fix them in the updated paper.

---

### Official Review · Reviewer_BHMR · 2021-07-15

**Rating:** 7
**Confidence:** 4

**Summary:**

In this paper, the authors study the gain in sample complexity of using kernel ridge regression with invariant versus non-invariant kernels. The authors consider a target function that is invariant through the action of a subgroup $G$ of the symmetric group in $d$ dimensions, and compare learning it with (1) an inner-product kernel (non-invariant) and (2) an inner-product kernel made invariant by group averaging over $G$. They then assume some classical capacity (on kernel (1)) and source (on the invariant target) conditions, and apply a standard upper bound on the generalization error of KRR. The main technical contribution is to notice that using a representation of projection (already noticed in a previous work [1]),  one can bound the degrees of freedom of the invariant kernel to get a factor $\nu_d (\ell_n)$ improvement in sample complexity with invariant kernels in the upper bound. This factor depends on the size of the group and some spectral properties of the group, and the authors provide upper bounds for several examples. Two-insights: the improvement factor can be up to the size of the group, and this does not break the curse of dimensionality.
The authors finally considered approximate invariance (that also commute with the polynomials) and get a similar upper bound, with a slightly looser factor.

**Limitations And Societal Impact:**

Yes

**Main Review:**

Exploiting invariance properties in the data is a central theme in machine learning, and is believed to play a major role in neural networks architecture for example. However, theoretical results are severely lacking and I find this work interesting and a great addition to the subject. The paper is well written and all the results are extensively discussed and interpreted.

However, here is a few things that would need some clarifications for me to raise my rating:

(1) Given the closeness to the paper [1], I would spend more time discussing and comparing the two works. Especially, the main insight of the proof (representation of projection) and the improvement factor were already claimed in [1], so I think the main contribution of this paper is to understand the dependency of this improvement factor on the structure of the group (in fixed $d$).
I understand that $d$ fixed and large $d$ are two different regimes with very different techniques/assumptions (source and capacity conditions for example). However I don’t think the high dimensional regime is too naive (learn only low-degree polynomials) as it simply confirm a series of (non-rigorous) work that show that RF/KRR act as shrinkage operators with an inflated effective regularization (which seems to be a good approximation even for moderate/low  $d$). In the case of invariant kernel, this effective regularization is divided by $d^\alpha$ and explain the improvement.

(2) As far I understand, [1] considers not only invariance by translation (as written in your introduction), but more generally, subsets of the orthogonal group O(d). This includes group with infinite size. In [1], the "degeneracy" of the group instead of the cardinality of the group controls the sample size gain (for a group of degeneracy $\alpha$, gain of factor $d^\alpha$ in sample complexity). So even for some infinite group, there is only a finite sample size gain. In some cases (some subgroups of permutation of polynomial size, while you can consider exponentially sized groups in your paper), the degeneracy coincides with the size of the group. While [1] fully check the conditions only for invariance by translation, I expect their proof to extend to these cases (with the added requirement $n > O(d^\alpha)$ if $\alpha >1$). I think it will be useful to discuss how the improvement factor differ in the two settings: how does size, spectral properties and degeneracy of the group influence the improvement factor for $d$ fixed and $d$ large ? It would be great to clarify this in the introduction.

(3) I would be more careful about claiming the factor improvement. A factor improvement in the generalization error upper bound does not necessarily imply a factor improvement in the actual generalization error. This would require to have a point wise lower bound with matching multiplicative constants (which is the case in the high-dimensional setting [1]). In particular, I am only aware of minimax lower bounds, but even in this case, the multiplicative constant can depend on the dimension. While I still believe these upper bounds are useful and match what happens in practice, I would add a paragraph discussing this point and different known lower bounds/or possible ways of obtaining them.


[1] Learning with invariances in random features and kernel models, Mei, Misiakiewicz, Montanari (2021).


---- Post-rebuttal ----

Thank you for the authors clarifications and for taking my suggestions. I am satisfied by the response and raised my rating to 7.

**Time Spent Reviewing:**

2

---

> ### Author Response · Authors · 2021-08-10
> **Response to Reviewer BHMR**
>
> Thank you for your insightful and thorough review.
>
> *Comparison to [1]*. Thank you for the important suggestion to compare the parameters of the problem across the two references (size and spectral statistics of the group in our case, degeneracy factor in their case). We fully agree with you that the high-dimensional regime studied by [1] is interesting and not "naive" (to the contrary, their characterization of what can be learned is very precise).
>
> Our main motivation for considering a different regime with fixed d, and $n \to \infty$ was that it seems to be more amenable to studying generic permutation groups, including large groups which may be of size exponential in d. In particular, while the results in [1] may plausibly be extended to groups of polynomial size in d, their analysis with polynomial scalings n ~ d^s is unlikely to extend to groups of exponential-in-d size (e.g., d^s divided by the group size would vanish as $d \to  \infty$), and to our understanding the current analysis only treats groups of size no more than d, rather than general polynomials. Nevertheless, the benefit of [1] is that the risk is characterized much more precisely for the specific groups and regimes they consider. We will provide a more thorough comparison to [1] in the final version of the paper, if it is accepted, clarifying the similarities and differences with our work.
>
> *Optimality.* Thank you for raising the issue of optimality. It is indeed true that the constants in the generalization bound may depend on dimension, making it challenging to argue for optimality. Nevertheless, given the constants from the assumptions (source, capacity and noise conditions), the bound does not depend on dimension other than through the invariance-related factor nu_d. We were not able to find lower bounds that show optimality of the dependence on such constants (the main lower bound we are aware of in this setting, due to Caponnetto and De Vito, only characterizes the lower rate, not the constant), but we are looking into studying the optimality of this constant for a longer version of the paper. We will discuss this issue further in the paper.

---

> ### Author Response · Authors · 2021-08-28
> **Update**
>
> Dear Reviewer BHMR,
>
> A few more remarks from our part:
>
>  * Using standard tools from minimax lower bounds, we were able to obtain a lower bound showing that the constant C4 (and its dependence on problem parameters) can be tight in a minimax sense for non-invariant targets in a RKHS ball, at least asymptotically (in n). You are right that what we'd really want is a pointwise bound for a fixed invariant target, however this may be difficult without a precise estimation of the risk (or at least we are unaware of how to obtain them), which is where the high-dimensional analyses a la [1] really shine.
>  * Regarding your second point, we realize that we had missed your remark on the possibility of group sizes larger than d^alpha. You are right to point this out, and we apologize for confusing group size and d^alpha in our previous comment. In this sense, an interesting question for future work would be to compare the gains in terms of degeneracy in such high-dimensional regimes with polynomial scalings, and group size in our asymptotic-in-n regime, for large groups.
>
> Thanks again for your insightful comments.
>
> Best,
> The authors

---

### Official Review · Reviewer_MSM8 · 2021-07-16

**Rating:** 6
**Confidence:** 3

**Summary:**

In the context of kernel ridge regression, the authors study the improvements in generalization error for estimating target functions that are invariant or stable under a subset of symmetry transformations, when using kernels that have the same invariance or stability properties. Specifically, the authors study estimation of functions on the d-dimensional sphere, and consider symmetry transformations given by permutations of the d coordinates. The main results are generalization error upper bounds in two contexts: (1) When the subset of symmetry transformations form a subgroup of the symmetric group, the target function is exactly invariant under this subgroup, and the invariant kernel is given by a symmetrization of an inner-product kernel---Theorem 5. (2) When the subset of symmetry transformations can be arbitrary subset of permutations, and both the target function and kernel are given by an averaging operation applied to an initial function and inner-product kernel, with respect to some distribution over the transformations of this subset---Theorem 13.

The improvement in generalization error for the invariant kernel over the initial inner-product kernel comes from a reduction in the degrees of freedom associated to the ridge regressor, which may be upper-bounded by comparing the dimension of invariant spherical harmonics with that of all spherical harmonics at a given frequency k. In the analyzed setting where symmetry transformations are |G| permutations of d coordinates, the authors show that this ratio of dimensions converges to 1/|G| in the limit of large frequency k -> infty, which is the relevant limit for large sample sizes n -> infty. The authors quantify this rate of convergence in k for a number of different examples of symmetry subgroups.

**Main Review:**

I think this paper explores an interesting question, and provides a concrete set of results that demonstrate the possible improvements of exploiting invariance in the kernel design. The extension in section 5 to stability rather than exact invariance also feels important from a practical perspective. Technically, the generalization error bounds in terms of N_{K_G} are perhaps not too surprising, and the more significant technical contributions might be in how to control N_{K_G} and the dimension ratios gamma_d, as carried out in Section 4 and Appendix C.

However, the insight provided by this paper also feels (to me) somewhat limited, for the following reasons:

(1) It's not clear how sharp are the obtained generalization bounds---in particular how sharp is the method of bounding N_{K_G} in Lemma 4, and the further restriction of l_n in Theorem 5 such that the second term of this bound dominates the first. I would have appreciated one or two concrete examples where these analyses may be shown to correctly characterize the size of N_{K_G}(lambda) for the relevant choice of lambda.

(2) As the authors recognize, since this dimension ratio nu_d in the risk bound (16) is lower-bounded by 1/|G|, the improvement in generalization error is not in the scaling dependence on n, but rather in how the "constant" depends on |G| and the dimension d. Thus the results are more significant for large subgroups or subsets |G|. I think this is fine, but then the analysis in Section 4 is a pointwise analysis over permutations in G, and thus doesn't really track well this dependence on d and |G|. For example, in the bound (17), nu_0 = 1/|G| can indeed be exponentially small in d, but can the leading constant C also be exponentially large in d? If n needs to be exponentially large in d for the statement "nu_d(l_n) is approximately 1/|G|" to hold, I think this detracts a bit from the relevance of the results.

-----------------------------------------

A couple other comments:

(3) In the proof of Theorem 13, can the authors perhaps also provide a proof of the analogue of Lemma 3 (the approximation error) in this setting? The result of Lemma 3 feels more intuitive to me in the setting of exact invariance and where S is a projection onto a subspace, but is less clear to me in the "geometric stability" setting of Section 5.

(4) Can the authors clarify what is the choice of h(sigma) in Proposition 14

**Time Spent Reviewing:**

4

---

> ### Author Response · Authors · 2021-08-10
> **Response to Reviewer MSM8**
>
> Thank you for your thorough and detailed review. Our replies follow.
>
> (1) While it may be possible to obtain sharper characterizations of N_{K_G}(lambda) which may jointly exploit decays of the eigenvalues mu_k and the ratios gamma_d(k), we chose to have a simpler upper bound that directly involves N_K(lambda). This allows us to have a final generalization bound that inherits properties of the original kernel while displaying the gains of invariance through nu_d(k), a quantity which we could control more easily, at least asymptotically for large k or n. Nevertheless, a sharper analysis of the degrees of freedom would be a promising direction.
>
> (2) Indeed, the constants in the asymptotic expansions of $\nu_d(\ell_n)$ may be large as a function of d, in particular they may depend both on properties of the group (i.e. the number of permutations in the group which have multiplicity leading to the exponent beta), but also on constants that come from the analysis of decay estimates, related to properties of the density q_sigma studied in the proof of Prop. 6. The dependence of these constants on d is thus quite involved, and we are planning to study this more precisely in a longer version of this work. Because of this, it may be easier to interpret our results through an "asymptotic in n" lens, where d is fixed and n grows to infinity, as is common in the nonparametric analyses of kernel methods, and here we also rely on such asymptotics for the improvement factor $\nu_d(\ell_n)$.
>
> (3) Thank you for pointing this out. In this case, there is not direct analog of Lemma 3, as the approximation error is not the same for the two kernels. We instead rely on the source condition (A5) for the invariant kernel for controlling the approximation error. We note that here the approximation error of the vanilla kernel can also be upper bounded using the same source condition, however the constant $C_{f^*}$ may be smaller in some situations — we will clarify this point, which was not clear in the current submission.
>
> (4) Thanks for raising this point, we will clarify that h is constant and equal to 1/|G|.

---

### Official Review · Reviewer_qjVL · 2021-07-20

**Rating:** 8
**Confidence:** 3

**Summary:**

This paper proposed a framework for analyzing the generalization behavior of kernel ridge regression under invariance and "near-invariance" assumptions with respect to group actions.

**Limitations And Societal Impact:**

- The numerical experiments are more qualitative than quantitative; the tightness of the established theoretical results are not thoroughly addressed.

- It's slightly confusing what the exponent $r$ means exactly in the notation $T^r_Kg$ in (A2). It can be guessed as a notation from functional calculus, but it would be good for the authors to confirm. It seems that (A2) is only used in the proof of Theorem 5 through reference [11].

- I have some confusion regarding how specific the established results depend on the specific group (permutation group) and domain (sphere). Consider a case where a group acts on the sphere continuously and transitively (e.g. Hopf fibration), so the target invariant function is effectively defined on a lower-dimensional quotient domain. This would change the dimension factor $d$ and constant $\nu_0$ simultaneously in e.g. (17). What do the theoretical results established in this paper look like on this "quotient" example (e.g. will they appear a bit conservative or pessimistic for this case)?

**Main Review:**

This is a highly inspirational and timely paper that shed new lights on learning under group actions.

Originality: The theoretical analysis under "geometric stability" in the context of kernel methods is unseen in previous literature, to the best of my knowledge. Technically, this work work is a novel combination of well-known techniques and new technical/combinatorial results. Related work are adequately cited.

Quality: The submission is technically sound to the best of my knowledge.

Clarity: The submission is clearly written and well organized.

Significance: The results are established under specific assumptions and relied upon particular properties of e.g. spherical harmonics. This imposes limits on the generality of the results, but the qualitative nature of the results (e.g. dependence of generalization error on the "effective dimension") can be expected to hold in much wider contexts.  important? It is very likely that the results in this paper will open the door for deeper explorations in this direction for other researchers/practitioners.


**Time Spent Reviewing:**

12 hours

---

> ### Author Response · Authors · 2021-08-10
> **Response to Reviewer qjVL**
>
> Thank you for the thorough and encouraging review. We address your comments below.
>
> *Generality, spherical data*. Our setting with data on the sphere was mainly chosen for theoretical convenience, as it is well suited for studying dot-product kernels, and we agree that it may provide a somewhat limited of invariance/stability in general. Establishing similar gains on more general data distributions is an important direction for future work.
>
> *Tightness.* This is a good point. Please see our response to Reviewer BHMR for more on this.
>
> *Exponent in (A2)*. The exponent refers to taking powers of a self-adjoint operator, which corresponds to taking powers of each eigenvalue while keeping eigenvectors fixed. We will clarify this notation in the text.
>
> *Role of group.* The questions of how to go beyond permutations or beyond spherical data are indeed important ones. Regarding the choice of group, we note that our analysis should readily extend to finite subgroups of the orthogonal group, even if this involves transformations that are not permutations. In particular, the dimension and the 1/|G| factor would remain the same in this case. What may change are the asymptotic decays in section 4, which will depend on spectral properties of the transformation matrices involved and may differ from those of permutations. We did not fully grasp the "quotient" example you suggested, but we hope this can provide a partial answer, and we are happy to discuss this further.

---

### Author Response · Authors · 2021-08-31
**Happy to clarify further**

Dear Reviewers,

We hope that our responses have properly addressed your questions and concerns. We understand that you may be overwhelmed with many other papers to review in your batch. If there are any concerns that could benefit from further clarifications, please let us know, we'd be happy to help.

Best,

The Authors

---

### Decision · Program_Chairs · 2021-09-27

**Decision:**

Accept (Poster)

**Comment:**

This paper studies kernel ridge regression in the case where the target function is invariant through the group action of $G$, a subgroup of the symmetric group. They explore how making the inner-product kernel invariant too can improve the sample complexity bounds. Furthermore they consider extensions to approximate invariance. One of the reviewers brought up the paper “Learning with invariances in random features and kernel models” by Mei et al. which has a number of overlapping messages. In their response, the authors clarified the technical relationship to this paper and the main differences are in the scaling regime and the types of permutation groups they can handle. Here they consider fixed $d$ and $n$ going to infinity, but can also work with large permutation groups whose size is exponential in $d$. This discussion should be incorporated into the next version of the paper. Overall the reviewers felt that this was a solid set of contributions on an important topic.